# Diffusion Fine-Tuning via Reparameterized Policy Gradient of the Soft Q-Function

**Hyeongyu Kang**[1] *   **Jaewoo Lee**[1][2] *   **Woocheol Shin**[1] *   **Kiyoung Om**[1]   **Jinkyoo Park**[1][3]

[1]KAIST   [2]MongooseAI   [3]Omelet

{khg2000v, jaewoo, woofe, se99an, jinkyoo.park}@kaist.ac.kr

## Abstract

Diffusion models excel at generating high-likelihood samples but often require alignment with downstream objectives. Existing fine-tuning methods for diffusion models significantly suffer from reward over-optimization, resulting in high-reward but unnatural samples and degraded diversity. To mitigate over-optimization, we propose **Soft Q-based Diffusion Finetuning (SQDF)**, a novel KL-regularized RL method for diffusion alignment that applies a reparameterized policy gradient of a training-free, differentiable estimation of the soft Q-function. SQDF is further enhanced with three innovations: a discount factor for proper credit assignment in the denoising process, the integration of consistency models to refine Q-function estimates, and the use of an off-policy replay buffer to improve mode coverage and manage the reward-diversity trade-off. Our experiments demonstrate that SQDF achieves superior target rewards while preserving diversity in text-to-image alignment. Furthermore, in online black-box optimization, SQDF attains high sample efficiency while maintaining naturalness and diversity. Our code is available at `https://github.com/Shin-woocheol/SQDF`.

## 1 Introduction

Diffusion models (Ho et al., 2020; Song et al., 2021b) have emerged as a dominant paradigm for generative tasks, achieving high-fidelity sample generation in domains such as text-to-image synthesis (Rombach et al., 2022), video generation (Ho et al., 2022), and biological molecules (Lee et al., 2025). This can be framed as an optimization problem: maximizing a reward signal that quantifies a desired property, such as aesthetic quality (Schuhmann, 2022), text-to-image alignment (Xu et al., 2023; Wu et al., 2023), or molecular bioactivity (Gosai et al., 2023).

Existing fine-tuning methods for reward optimization are broadly categorized into two main approaches: reinforcement learning (RL)-based methods (Black et al., 2023) and direct backpropagation methods (Xu et al., 2023; Clark et al., 2023; Prabhudesai et al., 2023). Although these methods effectively optimize rewards, their singular focus on reward maximization makes them highly prone to over-optimization (Skalse et al., 2022), often at the expense of sample quality and diversity. While KL-divergence regularization with pre-trained model is proposed to mitigate over-optimization (Uehara et al., 2024a), they often necessitate explicit value function training—a notoriously unstable process (Uehara et al., 2024b; Hu et al., 2025; Zhou et al., 2024)—or depend on high-variance Monte Carlo gradient estimators (Fan et al., 2023; Venkatraman et al., 2024). Although downstream reward gradients are powerful training signals, directly leveraging them to fine-tune diffusion models without over-optimization remains an open problem.

To this end, we propose **Soft Q-based Diffusion Finetuning (SQDF)**, a method that employs a reparameterized policy gradient guided by a training-free soft Q-function within a KL-regularized RL framework. The core of our approach is to approximate the soft Q-function via a single-step posterior mean approximation (Li et al., 2024), a strategy that circumvents the need for unstable value function learning. This approximation is differentiable under the parameterized oracle or proxy models (Xu et al., 2023; Wu et al., 2023; Uehara et al., 2024c), enabling the direct use of gradient for low-variance and sample-efficient policy updates (Haarnoja et al., 2018). The entire approach is

---

*Equal contribution authors.

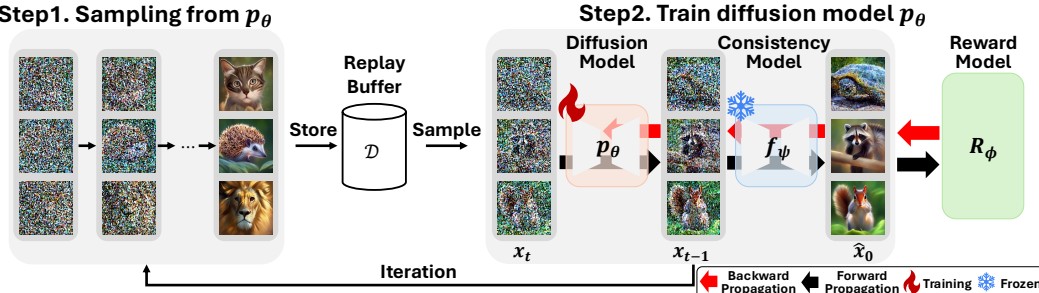

Figure 1: Overview of the SQDF framework. The process involves two stages: (1) samples generated by the diffusion model $p_\theta$ are stored in a replay buffer; (2) a noisy sample $x_t$ is drawn from the buffer and denoised one step by the diffusion model $p_\theta$. The consistency model $f_\psi$ then takes $x_{t-1}$ as input and predicts the clean sample $\hat{x}_0$. This prediction is evaluated by a reward model $r_\phi$, and the resulting reward gradient is used to update $p_\theta$ via a reparameterized policy gradient.

guided by a KL-regularized objective (Uehara et al., 2024c), which enforces the finetuned model to remain close to the pretrained data distribution, thereby preserving sample quality and diversity.

Beyond the foundation, SQDF introduces three additional innovations. (1) Incorporation of a discount factor $\gamma$ to downweight early denoising steps, reflecting their limited influence on final sample quality (Ho et al., 2020). (2) Integration of consistency models (Song et al., 2023), refining estimation quality of soft-Q function. (3) Off-policy updates with a replay buffer allow us to exploit its benefits for mode-coverage (Sendera et al., 2025), managing the reward-diversity trade-off by curating the data used for the finetuning of the diffusion model.

We evaluate SQDF in two settings. First, we apply SQDF to optimize Stable Diffusion 1.5 (Rombach et al., 2022) and Stable Diffusion XL (Podell et al., 2023) using differentiable rewards, specifically LAION aesthetic score (Schuhmann, 2022) and the HPSv2 human preference score (Wu et al., 2023). Our results demonstrate that SQDF effectively optimizes rewards while mitigating overoptimization. Second, we test SQDF in a black-box optimization problem, where optimizing reward and preserving diversity are both crucial under a limited query budget (Bengio et al., 2021). In this experiment, SQDF achieves high sample efficiency while preserving diversity.

## 2 RELATED WORKS

### 2.1 ENTROPY REGULARIZED REINFORCEMENT LEARNING

Entropy regularized RL augments the expected return with a Shannon entropy bonus or a KL divergence penalty. RL with Shannon entropy bonus, usually called as maximum entropy RL, enhances exploration and robustness, enabling deep RL methods to achieve strong performance in various domains (O'Donoghue et al., 2017; Haarnoja et al., 2017; 2018). KL-regularized RL encourages high-return actions while constraining the policy to remain close to a reference distribution (Wu et al., 2019; Nair et al., 2020). In SQDF, we employ a KL-regularized RL framework with a soft Q-function to constrain the policy to stay close to the pretrained diffusion model, thereby preserving naturalness and diversity of the pretrained diffusion model (Uehara et al., 2024a).

### 2.2 ALIGNMENT OF DIFFUSION MODELS FROM HUMAN FEEDBACK

The expressive capability of diffusion models in capturing data distributions is effective across various domains (Yang et al., 2023), but aligning pre-trained diffusion models with downstream objectives is often necessary. Black et al. (2023); Fan et al. (2023) employ PPO (Schulman et al., 2017b) to fine-tune the diffusion model by maximizing the black-box reward function. Other works (Xu et al., 2023; Prabhudesai et al., 2023; Clark et al., 2023) assume the reward gradient is accessible, and directly backpropagate gradient signals from the final state of the diffusion process.

While effective at optimizing rewards, they are prone to over-optimization, leading to semantic collapse and reduced diversity (Skalse et al., 2022; Gao et al., 2023). To address this issue, KL regularization approaches are introduced. While effective in mitigating over-optimization, Uehara et al. (2024b) requires value function training, notoriously hard to train in diffusion (Hu et al., 2025), as well as initial noise distribution training. Fan et al. (2023); Venkatraman et al. (2024) estimate the gradient signal with high-variance Monte Carlo estimation even if there exists a reward gradient signal available. Recently, Domingo-Enrich et al. (2025) introduce stochastic optimal control for finetuning flow-based models to reward tilted distribution. SQDF leverages a training-free, one-step soft Q-function approximation to directly apply the reward gradient via a reparameterized policy update, effectively fine-tuning the model without backpropagation through the denoising chain.

## 3 BACKGROUND

### 3.1 DIFFUSION MODELS

Diffusion model is a generative model that approximates the data distribution, generating samples by gradually denoising the Gaussian noise $x_T \sim \mathcal{N}(x_T; 0, I)$ via the parameterized reverse process $p_\theta(x_{0:T})$. The reverse process is defined as a Markov chain as follows:

$$p_\theta(x_{0:T}) = p(x_T) \prod_{t=1}^{T} p_\theta(x_{t-1} \,|\, x_t), \qquad p_\theta(x_{t-1} \,|\, x_t) = \mathcal{N}(x_{t-1}; \, \mu_\theta(x_t, t), \, \sigma_t^2 I) \qquad (1)$$

The forward process $q(x_{1:T}|x_0)$, is a Markov chain that gradually noises the $x_0$ from the data distribution according to the fixed variance schedule $\{\beta_t\}_{t=1}^{T}$.

$$q(x_{1:T} \,|\, x_0) = \prod_{t=1}^{T} \mathcal{N}(x_t; \, \sqrt{1-\beta_t} \, x_{t-1}, \, \beta_t I), \quad q(x_t|x_{t-1}) = \mathcal{N}(x_t; \, \sqrt{1-\beta_t} x_{t-1}, \, \beta_t I) \quad (2)$$

Diffusion models can be trained with the following noise prediction loss (Ho et al., 2020):

$$\mathcal{L}(\theta) = \mathbb{E}_{x_0 \sim p_0, \, t \sim \mathcal{U}\{1,...,T\}, \, \epsilon \sim \mathcal{N}(0,I)} \Big[ \big\| \epsilon - \epsilon_\theta \big( \sqrt{\bar{\alpha}_t} \, x_0 + \sqrt{1-\bar{\alpha}_t} \, \epsilon, \, t \big) \big\|_2^2 \Big] \qquad (3)$$

where $\bar{\alpha}_t = \prod_{s=1}^{t} (1 - \beta_s)$ with noise parameterization $\epsilon_\theta(x_t, t)$.

### 3.2 MARKOV DECISION PROCESS FOR DIFFUSION FINETUNING

The diffusion reverse process can be naturally formulated as a Markov Decision Process (MDP) due to its inherent Markov property. MDP is defined as $\mathcal{M} = (\mathcal{S}, \mathcal{A}, \mathcal{T}, \mathcal{R}, \rho_0)$, where $\mathcal{S}$ denotes the state space, $\mathcal{A}$ the action space, $\mathcal{T} : \mathcal{S} \times \mathcal{A} \rightarrow \mathcal{S}$ the transition function, $\mathcal{R} : \mathcal{S} \times \mathcal{A} \rightarrow \mathbb{R}$ the reward function, and $\rho_0$ the initial state distribution. We formulate a finite-horizon MDP with sparse rewards and the deterministic transition as follows:

$$s_t \triangleq (x_{T-t}, T-t) \quad \pi_\theta(a_t|s_t) \triangleq p_\theta(x_{T-t-1}|x_{T-t}) \quad P(s_{t+1}|s_t, a_t) \triangleq \delta_{(x_{T-t-1}, T-t-1)}$$

$$a_t \triangleq x_{T-t-1} \qquad\qquad \rho_0(s_0) \triangleq (\delta_T, \mathcal{N}(0, I)) \qquad\qquad R(s_t, a_t) \triangleq \begin{cases} r(x_0) & \text{if } t = T-1 \\ 0 & \text{otherwise} \end{cases}$$

### 3.3 KL-REGULARIZED RL IN DIFFUSION MDP

We define the KL-regularized RL objective for diffusion model alignment, using the MDP formulation in Section 3.2, following Uehara et al. (2024a):

$$p^* = \arg\max_{p_\theta} \mathbb{E}_{\tau \sim p_\theta(\tau)} \left[ r(x_0) - \alpha \sum_{t=1}^{T} \mathcal{D}_{KL}(p_\theta(\cdot \,|\, x_t) || p'(\cdot \,|\, x_t)) \right], \qquad (4)$$

where $\tau = (x_T, x_{T-1}, ..., x_1, x_0)$ denotes denoising trajectory generated by $p_\theta$, and $p'$ denotes pretrained diffusion model as reference policy. If we define the optimal soft Q-function for a state-action pair $(x_t, x_{t-1})$ as the sum of reward with KL regularization:

$$Q^*_{\text{soft}}(x_t, x_{t-1}) = \mathbb{E}_{p^*} \left[ r(x_0) - \alpha \sum_{k=1}^{t-1} D_{KL}(p^*(\cdot|x_k) || p'(\cdot|x_k))) \Big| x_t, x_{t-1} \right], \qquad (5)$$

then we can analytically derive the corresponding optimal policy:

$$p^*(\cdot|x_t) = \underset{p_\theta(\cdot|x_t)}{\arg\max}\, \mathbb{E}_{x_{t-1}\sim p_\theta(\cdot|x_t)}\left[Q^*_{\text{soft}}(x_t, x_{t-1}) - \alpha D_{\text{KL}}(p_\theta(\cdot|x_t)||p'(\cdot|x_t))\right]. \quad (6)$$

To obtain the soft optimal policy in Equation (6), we need the optimal soft Q-function which is characterized by the soft Bellman equation:

$$Q^*_{\text{soft}}(x_t, x_{t-1}) = R(x_t, x_{t-1}) + V^*_{\text{soft}}(x_{t-1}), \quad (7)$$

$$V^*_{\text{soft}}(x_t) = \alpha \log \mathbb{E}_{x_{t-1}\sim p'(\cdot|x_t)}\left[\exp\left(\frac{R(x_t, x_{t-1}) + V^*_{\text{soft}}(x_{t-1})}{\alpha}\right)\right]. \quad (8)$$

After recursively solving the soft Bellman equation, the soft Q-function can be approximated as:

$$Q^*_{\text{soft}}(x_t, x_{t-1}) = \alpha \log \mathbb{E}_{x_0,\ldots,x_{t-2}\sim p'(\cdot|x_{t-1})}\left[\exp\left(\frac{r(x_0)}{\alpha}\right)\Big|x_{t-1}\right] \approx r(\hat{x}_0(x_{t-1})), \quad (9)$$

where $\hat{x}_0(x_t) = \mathbb{E}_{p'}[x_0|x_t]$ is the posterior mean approximation driven from Tweedie's formula (Efron, 2011; Chung et al., 2023). We provide details in Section A.

## 4 METHODS

This section details our proposed method, SQDF, a novel KL-regularized RL method for fine-tuning diffusion models while mitigating reward over-optimization. The core of SQDF lies in bridging the signal from the reward model directly to the intermediate denoising process of the diffusion through the training-free soft Q-function. To enhance the stability and effectiveness, we introduced three components. (1) discount factor $\gamma$, (2) consistency model, and (3) experience replay buffer. We illustrate the overview of our method in Figure 1.

### 4.1 POLICY IMPROVEMENT VIA REPARAMETERIZED POLICY GRADIENT

Gradients from a differentiable reward function offer a potent signal for fine-tuning diffusion models (Clark et al., 2023; Xu et al., 2023), yet existing KL-regularized RL methods struggle to leverage them directly (Uehara et al., 2024b; Venkatraman et al., 2024). SQDF directly exploits the reward gradient to serve as the gradient of the soft Q-function, approximated in a training-free manner in Equation (9). This approach eliminates the need to train a separate Q-network and avoids its associated instabilities (Zhou et al., 2024; Hu et al., 2025).

Leveraging the reward gradient as the Q-function gradient allows us to employ the reparameterized policy gradient, which provides a low-variance and accurate training signal for reward optimization. By substituting the soft Q-function in Equation (6) with the soft Q-function approximation in Equation (9), we can derive a reparameterized policy gradient loss with the reward function:

$$\mathcal{L}(\theta) = \mathbb{E}_{x_t}\left[\mathbb{E}_{x_{t-1}\sim p_\theta(\cdot|x_t)}[-Q^*_{\text{soft}}(x_t, x_{t-1}) + \alpha D_{\text{KL}}(p_\theta(x_{t-1}|x_t)||p'(x_{t-1}|x_t))]\right] \quad (10)$$

$$\approx \mathbb{E}_{x_t}\left[\mathbb{E}_{x_{t-1}\sim p_\theta(\cdot|x_t)}[-r(\hat{x}_0(x_{t-1})) + \alpha D_{\text{KL}}(p_\theta(x_{t-1}|x_t)||p'(x_{t-1}|x_t))]\right]. \quad (11)$$

Since $x_{t-1}$ sampled from $p_\theta(\cdot|x_t)$ is a stochastic variable, the gradient from $r(\hat{x}_0)$ cannot be back-propagated. Therefore, by using reparameterization with $x_{t-1} = \mu_\theta(x_t, t) + \sigma_t\epsilon$ (Kingma & Welling, 2013), we utilize the gradient signals can be utilized to update the policy parameters (Lillicrap et al., 2015; Haarnoja et al., 2018). Then the gradient of the Equation (11) is given by:

$$\nabla_\theta \mathcal{L}(\theta) = \mathbb{E}_{x_t}\left[\mathbb{E}_{\epsilon\sim\mathcal{N}(0,I)}\left[-\nabla_{x_{t-1}}r(\hat{x}_0(x_{t-1}))\cdot\nabla_\theta\mu_\theta(x_t, t) + \alpha\nabla_\theta D_{KL}(p_\theta||p')\right]\right]. \quad (12)$$

### 4.2 STABILIZATION AND OFF-POLICY TECHNIQUES IN SQDF

Although the reparameterized policy gradient in Equation (10) directly leverages the powerful reward gradient signal, it still faces significant challenges. In the early denoising steps, due to a lower signal-to-noise ratio (Kingma et al., 2021), each denoising step has a minimal influence on the final generated samples (Ho et al., 2020). Moreover, approximation error from Equation (9) exacerbates in early denoising steps, which leads to unreliable training siganl. To address these issues, SQDF incorporates two stabilization techniques: the adoption of a discount factor $\gamma$ and consistency models. Separately, with the inherent off-policy training of Equation (11), we utilize buffer $\mathcal{D}$ to control reward-diversity trade-offs.

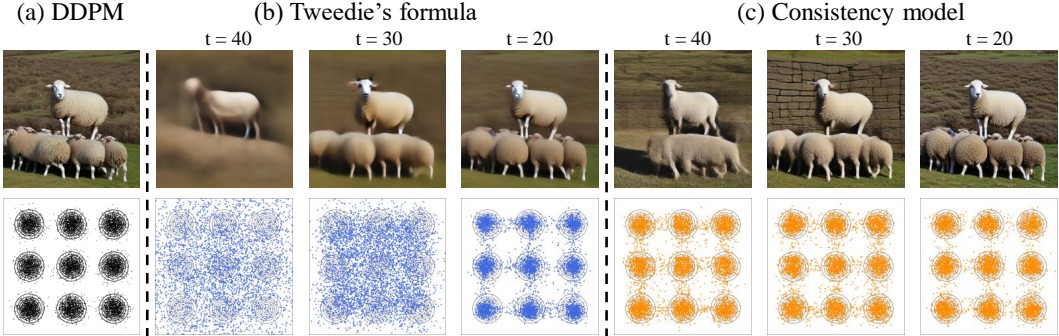

Figure 2: Comparison of multi-step sampling with one-step $x_0$ estimation. (a): DDPM 50-step sampling accurately capture $x_0$ distribution. (b): A one-step $x_0$ estimation via Tweedie's formula is highly inaccurate, particularly at early denoising steps. (c): Consistency model, however, provides an $x_0$ estimate with uniform accuracy across all timesteps.

### 4.2.1 $\gamma$ DISCOUNTED MDP FOR IMPROVED CREDIT ASSIGNMENT

Due to inherent stochasticity in the diffusion reverse process, the influence of action at $x_t$ on the final sample $x_0$ diminishes as $t$ increases (Ho et al., 2020; Song et al., 2021b). Prior MDP formulation (Black et al., 2023) typically sets $\gamma = 1$, assigning equal credits to actions taken across all timesteps. For better credit assignment, we introduce a discount factor $\gamma \in [0,1]$. By setting $\gamma < 1$, it scales the credit for an action at timestep $t$ by $\gamma^t$, exponentially down-weighting actions in early denoising steps in large $t$ to align the optimization with their limited contribution to the final sample. Then we can rewrite the KL-regularized RL objective at Equation (4) with discount factor $\gamma$ as follows:

$$p^* = \arg\max_{p_\theta} \mathbb{E}_{\tau \sim p_\theta(\tau)} \left[ \gamma^{T-1} r(x_0) - \alpha \sum_{t=1}^{T} \gamma^{T-t} \mathcal{D}_{KL}(p_\theta(\cdot \mid x_t) \| p'(\cdot \mid x_t)) \right] \tag{13}$$

In the case of discounted MDP, a recursive expansion of the soft optimal value function denoted at Equation (8) leads to the following bounds for $Q^*_{\text{soft}}(x_t, x_{t-1})$:

$$\alpha \log \mathbb{E}_{x_{0:t-2} \sim p'(\cdot \mid x_{t-1})} \left[ \exp\left( \frac{\gamma^{t-1}}{\alpha} r(x_0) \right) \right] \leq Q^*_{\text{soft}}(x_t, x_{t-1}) \leq \gamma^{t-1} \alpha \log \mathbb{E}_{x_{0:t-2} \sim p'(\cdot \mid x_{t-1})} \left[ \exp\left( \frac{r(x_0)}{\alpha} \right) \right] \tag{14}$$

From the above inequalities, we observe that applying posterior mean approximation to both the lower and upper bounds of the soft Q-function yields the identical first-order approximation. Motivated by this observation, we approximate the soft Q-function under the discounted MDP as:

$$Q^*_{\text{soft}}(x_t, x_{t-1}) \approx \gamma^{t-1} r(\hat{x}_0(x_{t-1})). \tag{15}$$

The detailed derivation of the above equations can be found in Section B.

### 4.2.2 CONSISTENCY MODEL FOR BETTER SOFT Q ESTIMATION

The critical limitation of Tweedie's formula for approximating the posterior mean $\hat{x}_0$ is its lack of reliability under high noise levels (Chung et al., 2023). This issue is illustrated in Figure 2-(b), where Tweedie's formula often yields samples that lie far outside the data distribution with the early diffusion timesteps. Although using a discount factor $\gamma$ can mitigate the effects of an inaccurate approximation, it can still mislead the policy update process (Jain et al., 2025). Although employing multi-step Ordinary Differential Equation (ODE) sampling (Song et al., 2021a) is suitable for accurate mean posterior approximation, it is impractical due to the requirement of backpropagation through the denoising trajectory, which leads to gradient instability (Clark et al., 2023).

To address these limitations, we employ the consistency model (Song et al., 2023) $f_\psi(x_t)$ to improve the accuracy of the posterior mean approximation. Consistency models are trained to map the noisy input $x_t$ to the corresponding clean image $x_0$ by distilling integration of probability flow ODE (Lu & Song, 2024). By using the consistency model as a reference diffusion model $p'$, we achieve a more accurate posterior mean than that provided by Tweedie's formula. As shown in Figure 2-(c), the consistency model provides a better $x_0$ approximation. We argue that accurate posterior mean prediction improves the soft-Q function approximation and better guides training.(See Section 5.3.)

### 4.2.3 OFF-POLICY TRAINING WITH REPLAY BUFFER

Unlike previous works that rely on on-policy samples (Black et al., 2023; Clark et al., 2023; Domingo-Enrich et al., 2025), the proposed SQDF loss in Equation (10) naturally accommodates off-policy updates. We integrate a replay buffer, denoted as $\mathcal{D}$, along with an experience replay strategy explained in Section E.1. This approach allows for the reuse of rare, high-reward, and diverse samples, facilitating a balance in the reward-diversity trade-off and enhancing mode coverage. Finally, combining all the key factors of the SQDF, we can define SQDF loss as follows:

$$\mathcal{L}_{\text{SQDF}}(\theta) = \mathbb{E}_{x_t \sim \mathcal{D},\ x_{t-1} \sim p_\theta}[-\gamma^{t-1} r\left(f_\psi\left(x_{t-1}\right)\right) + \alpha D_{\text{KL}}(p_\theta(x_{t-1}|x_t)||p'(x_{t-1}|x_t))]. \quad (16)$$

## 5 EXPERIMENTS

This section presents a comprehensive empirical analysis of SQDF. For all tasks, we utilize Stable Diffusion v1.5 (Rombach et al., 2022), and all experiments are conducted using three random seeds. Our experiments aim to answer the following research questions:

- Does SQDF optimize the reward while mitigating over-optimization? (§5.1)
- Can SQDF effectively finetune diffusion models in online black-box optimization settings? (§5.2)
- How do the individual components of SQDF contribute to improved finetuning performance in terms of efficiency, diversity, and stability? (§5.3)

### 5.1 FINETUNING FOR TEXT-TO-IMAGE DIFFUSION MODELS

For finetuning the text-to-image diffusion models, we employ the LAION aesthetic predictor (Schuhmann, 2022) and human preference scores (HPSv2) (Wu et al., 2023) as objective functions. Additional details of the task and algorithms can be found in Section D.2.2 and Algorithm 1.

### 5.1.1 EXPERIMENT SETUP

**Evaluation metrics:** As shown in prior works (Black et al., 2023; Clark et al., 2023), reward over-optimization often leads to unrecognizable semantic content and reduced diversity. We refer to these two phenomena as *semantic collapse* and *diversity collapse* respectively. To assess semantic collapse, we employ the prompt alignment scores such as HPSv2 (Wu et al., 2023), ImageReward (Xu et al., 2023), and PickScore (Kirstain et al., 2023), which are trained to align text-image pairs with human preference datasets. To detect diversity collapse, we employ two diversity measures: the mean pairwise distance computed with LPIPS (Zhang et al., 2018) and the mean pairwise cosine distance calculated using DreamSim features (Fu et al., 2023). We provide visual examples of semantic and diversity collapse cases in Section C.

**Baselines:** We compare against: (1) RL-based method: DDPO (Black et al., 2023), (2) direct back-propagation method: DRaFT (Clark et al., 2023), ReFL (Xu et al., 2023), and (3) KL-regularized variants of DDPO and DRaFT, which introduce an auxiliary KL-regularization term in the reward. Further details of KL-regularized baselines are provided in Section D.

### 5.1.2 RESULTS

**SQDF mitigates over-optimization:** Figure 3 shows how alignment and diversity scores change as the target reward is optimized. The first row demonstrates the optimization of the aesthetic score, and the second row illustrates the HPS optimization. Panels (a)–(d) show that gradient-based baselines such as ReFL and DRaFT achieve high aesthetic scores but suffer sharp declines in alignment and diversity, highlighting their vulnerability to reward over-optimization. DDPO, which does not exploit gradient signals, fails to reach a comparable aesthetic score and exhibits rapid diversity collapse. For optimizing HPS score, (e)–(h) demonstrate that SQDF consistently achieves the highest alignment and diversity at equivalent reward levels, demonstrating its generalization capabilities.

**Comparison with KL-Augmented Baselines:** A natural question is whether the benefits of SQDF can be replicated by simply adding a KL-divergence term to existing methods. To address this, we compare the trade-off curves generated by sweeping the regularization strength $\alpha$ for both SQDF and other KL-augmented baselines. The results, depicted in Figure 4, show that SQDF achieves higher

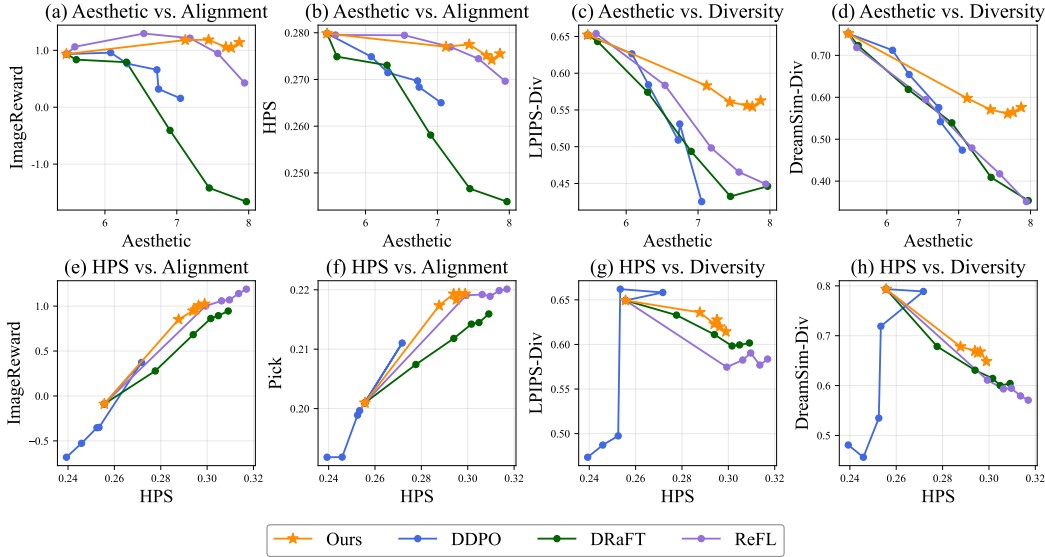

Figure 3: Comparison of evaluation metrics during optimization of the target reward. Top: The target reward is the LAION aesthetic score. Bottom: The target reward is HPSv2. (a), (b), (e), and (f): evaluation of alignment score using ImageReward and HPS. (c), (d), (g), and (h): evaluation of diversity using LPIPS and DreamSim.

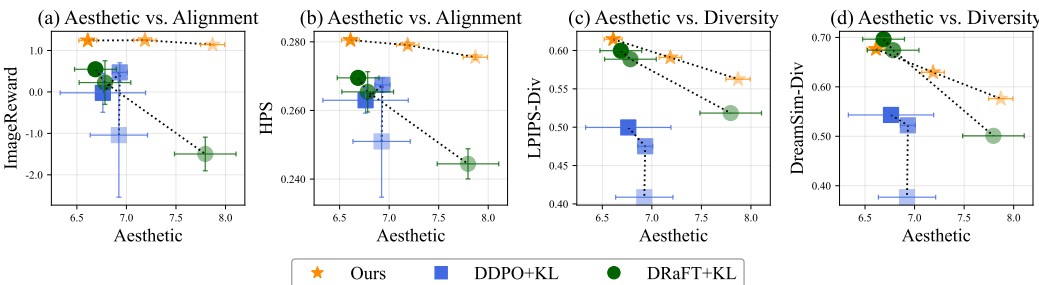

Figure 4: Comparison of trade-off curves with KL-regularized baselines. Curves are obtained by varying the KL-regularization coefficient $\alpha$. Darker points correspond to a stronger KL-regularizer.

rewards while maintaining better alignment and diversity, mostly occupying the Pareto optimality across different metrics. These findings suggest that SQDF provides a more effective framework for optimizing the KL-regularized objective with reward gradient backpropagation while mitigating over-optimization. Qualitative comparisons can be found in Figure 5 and Section E.

## 5.2 ONLINE BLACK-BOX OPTIMIZATION WITH DIFFUSION MODELS

Given a limited query budget for evaluating an oracle reward function, online black-box optimization as suggested by Uehara et al. (2024c) involves fine-tuning a generative model to achieve high oracle reward scores while preserving the model's naturalness and diversity. In this experiment, we adopt the experimental setting from (Uehara et al., 2024c), iteratively fine-tune a diffusion model with aesthetic score considered as a black-box oracle function. Additional details of the task and algorithms can be found in Section D.2.3 and Algorithm 2.

### 5.2.1 EXPERIMENT SETUP

**Baselines:** We compare **SQDF** against: **(1) SEIKO** (Uehara et al., 2024c), a KL-regularized direct backpropagation approach that utilizes the reward signal of the proxy reward model. We evaluate two variants: SEIKO-UCB and SEIKO-Bootstrap, which introduce an uncertainty bonus to improve

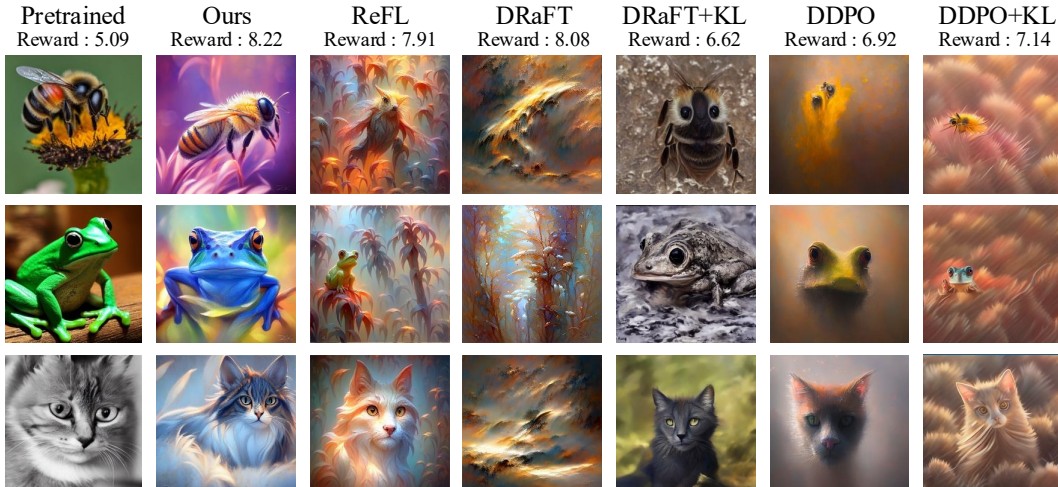

| Pretrained | Ours | ReFL | DRaFT | DRaFT+KL | DDPO | DDPO+KL |
|:---:|:---:|:---:|:---:|:---:|:---:|:---:|
| Reward : 5.09 | Reward : 8.22 | Reward : 7.91 | Reward : 8.08 | Reward : 6.62 | Reward : 6.92 | Reward : 7.14 |

Figure 5: Comparison of generated images from different fine-tuning methods, using model checkpoints selected when a reward of 8.0 was achieved (or the maximum reward if 8.0 was not reached). The average reward for the presented images is shown for each method.

| Method | Target (Aesthetic) | Alignment | | Diversity | |
|---|---|---|---|---|---|
| | | (ImageReward) | (Hps) | (Lpsis-Div) | (Dreamsim-Div) |
| PPO+KL | 6.63 (0.45) | -1.35 (1.30) | 0.24 (0.01) | 0.47 (0.07) | 0.44 (0.15) |
| SEIKO-Bootstrap | 7.80 (0.11) | -1.69 (0.64) | 0.23 (0.01) | 0.36 (0.08) | 0.24 (0.13) |
| SEIKO-UCB | 7.49 (0.18) | -1.08 (0.96) | 0.24 (0.01) | 0.40 (0.12) | 0.32 (0.24) |
| SQDF-Bootstrap | **7.87 (0.11)** | **1.14 (0.08)** | **0.27** (0.00) | 0.49 (0.02) | 0.53 (0.02) |
| SQDF-UCB | **7.87 (0.15)** | 1.10 (0.12) | **0.27** (0.00) | **0.51 (0.03)** | **0.54 (0.04)** |

Table 1: Results of online black-box optimization. SQDF achieves superior performance under the same oracle query budgets. We report the mean and standard deviation over 3 random seeds.

exploration. **(2) PPO + KL**, where the policy is fine-tuned directly with oracle rewards via a KL-penalized PPO update (Schulman et al., 2017b).

**Training surrogates and uncertainty bonus:** We train our proxy of the reward model and introduce an uncertainty bonus following SEIKO. Detailed in Section D.1.2.

### 5.2.2 RESULTS

Table 1 reveals clear superiority of SQDF across all evaluation metrics. While SQDF consistently achieves the highest target rewards alongside strong alignment and diversity scores, SEIKO, the direct propagation method, suffers from a critical trade-off: both variants show deteriorating alignment and reduced diversity as optimization progresses. This stark contrast demonstrates that even under an identical KL regularization framework, our approach fundamentally outperforms existing methods. Unlike controlled fine-tuning tasks, this MBO setting with imperfect reward proxies typically drives models out of distribution, losing the naturalness. Interestingly, SQDF maintains robustness across all objectives simultaneously, proving its robustness to inaccurate rewards. Figure 11 offers visual comparison between SEIKO and SQDF.

### 5.3 ABLATION STUDY

To provide deeper insights into SQDF's effectiveness, we present comprehensive ablation studies examining each of the three key techniques outlined in Section 4.2.

**SQDF component ablation:** First, we examine the effect of the discount factor $\gamma$. In the Equation (13), prior works implicitly use $\gamma = 1$, whereas our setting adopts $\gamma \in [0, 1)$. As shown in Figure 6, removing the discount factor ($\gamma = 1$) ultimately achieves a higher aesthetic score. How-

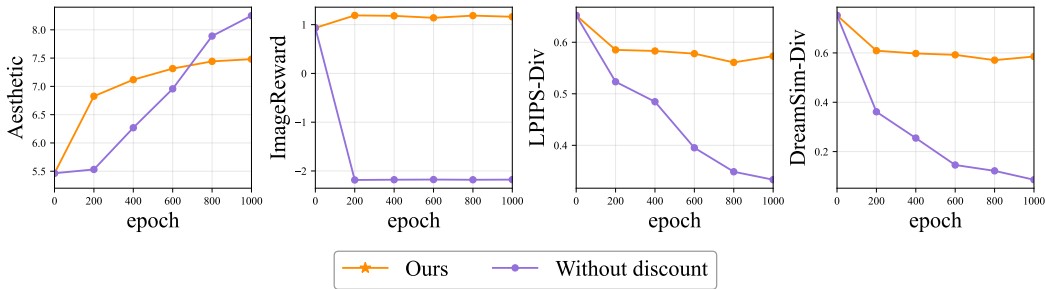

Figure 6: The effect of the discount factor $\gamma$ on training dynamics. Comparison of our method against a baseline where the discount factor is set to $\gamma = 1$.

ever, undiscounted SQDF exhibits slower optimization at early epochs and a significant drop in alignment and diversity scores. This suggests that introducing small credit near the initial state of the diffusion process improves credit assignment and reduces the high approximation error in the early denoising steps. We report the result of varying $\gamma$ in Section E.2.

Table 2 presents the performance of SQDF when the consistency model or replay buffer is ablated. Removing the consistency model decreases performance on the target reward. This indicates that the use of the consistency model improves training efficiency by making the soft Q-function approximation more reliable, as discussed in Section 4.2.2. Removing the buffer reduces diversity scores, as also shown in Section E.1, suggesting that the buffer helps preserve model support by reusing past experience. This aligns with prior findings that replay buffers improve mode coverage and mitigate catastrophic forgetting (Mnih et al., 2013).

|  | Evaluation Metric | | |
|---|---|---|---|
| **Method** | Aesthetic | DreamSim-Div | LPIPS-Div |
| SQDF | 7.87 | 0.58 | 0.56 |
| w/o CM | 7.10 | 0.62 | 0.59 |
| w/o buffer | 8.06 | 0.56 | 0.55 |

Table 2: Ablation study on Consistency model (CM) and Buffer. The consistency model enables faster convergence, while the buffer preserves diversity.

**DDIM vs consistency:** The naive one-step prediction of $\hat{x}_0$ using Tweedie's formula is highly inaccurate in the early denoising steps, as illustrated in Figure 2-(b). A natural alternative is to use $n$-step ODE sampling with DDIM (Song et al., 2021a) for accurate approximation. As shown in Figure 7, SQDF with 2-step DDIM sampling substantially improves optimization compared to Tweedie's formula. However, 4-step DDIM leads to unstable training, which we attribute to high variance in the gradients. We adopt the consistency model as a Pareto solution for reliable $x_0$ prediction and stable training.

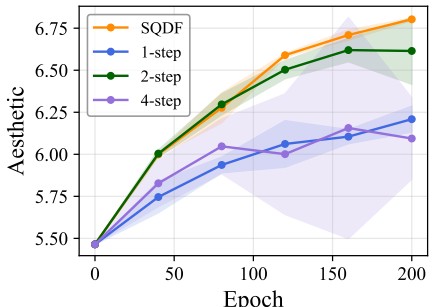

Figure 7: DDIM $n$-step versus consistency model as a $\hat{x}_0$ predictor.

## 6 CONCLUSION

In this work, we introduce SQDF, a novel KL-regularized reinforcement learning framework for fine-tuning diffusion models that directly utilizes reward gradients. The core of our method is a reparameterized policy gradient with a training-free, one-step approximation of the soft Q-function. This allows SQDF to efficiently optimize for downstream objectives while effectively mitigating reward over-optimization. We further enhance training stability by incorporating three key components: a discount factor $\gamma$ for credit assignment, a consistency model for reliable soft Q estimation, and a replay buffer to manage the reward-diversity trade-off. Extensive experiments demonstrate that SQDF successfully optimizes target rewards while preserving naturalness and diversity, thereby pushing the Pareto frontier in both text-to-image fine-tuning and black-box optimization tasks. We believe that leveraging more advanced one-step distillation models and sophisticated buffer management techniques are promising directions for future work.

## ACKNOWLEDGEMENT

We thank the anonymous reviewers for their insightful comments and suggestions, which significantly improve our manuscript. This work is supported by the National Research Foundation of Korea (NRF) grant funded by the Korea government (MSIT)(No.RS-2022-NR068758), NRF grant funded by the Korea government (MSIT) (No. RS-2024-00410082), and NRF grant funded by the Korea government(MSIT) (No. RS-2025-00563763).

## THE USE OF LARGE LANGUAGE MODELS

Large Language Models were employed exclusively for two auxiliary tasks: (1) minor polishing of the manuscript text for improving grammar and readability, and (2) limited assistance in code implementation for debugging syntax or refactoring functions. Importantly, LLMs did not contribute to the conception of the research problem, the development of the core methodology, or the design and execution of the experiments. All critical ideas, methods, and analyses presented in this paper are the original work of the authors.

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

# A KL-REGULARIZED RL IN DIFFUSION MDP

## A.1 DERIVATION OF SOFT OPTIMAL POLICY

In this section, we present the derivation of the soft optimal policy in KL-regularized RL for diffusion MDP in Section 3.2. Since we aim to maximize rewards while mitigating over-optimization, we define the objective as follows (Uehara et al., 2024a):

$$p^* = \arg\max_{p_\theta} \mathbb{E}_{\tau \sim p_\theta(\tau)} \left[ \sum_{t=1}^T \Big( R(x_t, x_{t-1}) - \alpha \mathcal{D}_{KL}(p_\theta(\cdot \mid x_t) || p'(\cdot \mid x_t)) \Big) \right] \quad (17)$$

$$= \arg\max_{p_\theta} \mathbb{E}_{\tau \sim p_\theta(\tau)} \left[ r(x_0) - \alpha \sum_{t=1}^T \mathcal{D}_{KL}(p_\theta(\cdot \mid x_t) || p'(\cdot \mid x_t)) \right], \quad (18)$$

where $\tau = (x_T, ..., x_0)$ denotes the trajectory denoised by training policy $p_\theta$ and $p'$ denotes reference policy (i.e. pre-trained diffusion model). So diffusion model $p_\theta$ tries to maximize reward while staying close to the pre-trained diffusion model, which has naturalness and diversity. Then we define the soft Q-function $Q_{\text{soft}}^{p_\theta}(x_t, x_{t-1})$ as the expectation of reward with the KL-divergence through the trajectory from the state-action pair $(x_t, x_{t-1})$:

$$Q_{\text{soft}}^{p_\theta}(x_t, x_{t-1}) = R(x_t, x_{t-1}) + \mathbb{E}_{p_\theta} \left[ \sum_{k=1}^{t-1} \Big( R(x_t, x_{t-1}) - \alpha \mathcal{D}_{KL}(p_\theta(\cdot \mid x_t) || p'(\cdot \mid x_t)) \Big) \Big| x_t, x_{t-1} \right] \quad (19)$$

$$= \mathbb{E}_{p_\theta} \left[ r(x_0) - \alpha \sum_{k=1}^{t-1} D_{KL}(p_\theta(\cdot | x_k) || p'(\cdot | x_k)) \Big| x_t, x_{t-1} \right]. \quad (20)$$

By using the definition of soft Q-function, we can reformulate our objective in Equation (18) as follows (Haarnoja et al., 2017):

$$p^* = \arg\max_{p_\theta} \sum_{t=1}^T \mathbb{E}_{x_t} \left[ \mathbb{E}_{x_{t-1} \sim p_\theta(\cdot | x_t)} [Q_{\text{soft}}^{p_\theta}(x_t, x_{t-1})] - \alpha D_{KL}(p_\theta(\cdot | x_t) || p'(\cdot | x_t)) \right] \quad (21)$$

We can define soft optimal Q function $Q_{\text{soft}}^*(x_t, x_{t-1})$ as expected return of taking action $x_{t-1}$ from state $x_t$ and subsequently following the optimal policy $p^*$:

$$Q_{\text{soft}}^*(x_t, x_{t-1}) = \mathbb{E}_{x_{0:t-1} \sim p^*(x_{0:t-1}|x_t)} \left[ r(x_0) - \alpha \sum_{k=1}^{t-1} D_{KL}(p^*(\cdot | x_k) || p'(\cdot | x_k)) \Big| x_t, x_{t-1} \right]. \quad (22)$$

By replacing the soft optimal Q function 22 with Equation (21), our objective simplifies to finding the policy $p_\theta(\cdot | x_t)$ that maximizes the expected soft optimal Q-function while remaining close to the reference policy $p'$:

$$p^*(\cdot | x_t) = \arg\max_{p_\theta} \mathbb{E}_{x_{t-1} \sim p_\theta(\cdot | x_t)} [Q_{\text{soft}}^*(x_t, x_{t-1}) - \alpha D_{KL}(p_\theta(\cdot | x_t) || p'(\cdot | x_t)) | x_t] \quad (23)$$

Then by expanding the Equation (23) explicitly, we can re-express the objective maximization problem as a KL divergence minimization problem (Haarnoja et al., 2018):

$$p^*(\cdot \mid x_t) = \arg\max_{p_\theta(\cdot|x_t)} \mathbb{E}_{x_{t-1} \sim p_\theta(\cdot|x_t)} \left[ Q_{\text{soft}}^*(x_t, x_{t-1}) - \alpha \log \frac{p_\theta(x_{t-1} \mid x_t)}{p'(x_{t-1} \mid x_t)} \right] \quad (24)$$

$$= \arg\max_{p_\theta(\cdot|x_t)} \mathbb{E}_{x_{t-1} \sim p_\theta(\cdot|x_t)} \left[ \log \frac{p'(x_{t-1} \mid x_t) \exp\big(Q_{\text{soft}}^*(x_t, x_{t-1})/\alpha\big)}{p_\theta(x_{t-1} \mid x_t)} \right] \quad (25)$$

$$= \arg\min_{p_\theta(\cdot|x_t)} D_{KL}\bigg( p_\theta(\cdot \mid x_t) \bigg\| \frac{1}{Z(x_t)} p'(\cdot \mid x_t) \exp\big(\tfrac{1}{\alpha} Q_{\text{soft}}^*(x_t, \cdot)\big) \bigg), \quad (26)$$

where $Z(x_t) = \int p'(x_{t-1}|x_t) \exp(Q_{soft}^*(x_t, x_{t-1})/\alpha) dx_{t-1}$ is a partition function which normalizes the distribution. Then the closed form soft optimal policy $p^*(\cdot|x_t)$ is given by:

$$p^*(\cdot \mid x_t) = \frac{p'(\cdot \mid x_t) \exp\big(Q_{\text{soft}}^*(x_t, \cdot)/\alpha\big)}{\int p'(x_{t-1} \mid x_t) \exp\big(Q_{\text{soft}}^*(x_t, x_{t-1})/\alpha\big) dx_{t-1}} \quad (27)$$

## A.2 SOFT BELLMAN EQUATION

We begin by defining the soft optimal value function as follows:

$$V_{\text{soft}}^*(x_t) = \mathbb{E}_{p^*}\left[\sum_{k=1}^{t}\Big(R(x_k, x_{k-1}) - \alpha D_{KL}(p^*(\cdot|x_k)||p'(\cdot|x_k))\Big)\Big|x_t\right] \tag{28}$$

$$= \mathbb{E}_{p^*}\left[r(x_0) - \alpha \sum_{k=1}^{t} D_{KL}(p^*(\cdot|x_k)||p'(\cdot|x_k)))\Big|x_t\right]. \tag{29}$$

Then we can express the soft optimal Q-function in terms of the soft value function:

$$Q_{\text{soft}}^*(x_t, x_{t-1}) = R(x_t, x_{t-1}) + \mathbb{E}_{p^*}\left[\sum_{k=1}^{t-1}\Big(R(x_k, x_{k-1}) - \alpha D_{KL}(p^*(\cdot|x_k)||p'(\cdot|x_k))\Big)\Big|x_t, x_{t-1}\right]$$

$$= R(x_t, x_{t-1}) + V_{\text{soft}}^*(x_{t-1}) \tag{30}$$

In the same way, we can rewrite the soft optimal value function as the soft optimal Q-function and the KL-divergence at the current state $x_t$:

$$V_{\text{soft}}^*(x_t) = \mathbb{E}_{x_{t-1}\sim p^*(\cdot|x_t)}\left[\mathbb{E}_{p^*}\left[\sum_{k=1}^{t}\Big(R(x_k, x_{k-1}) - \alpha D_{KL}(p^*(\cdot|x_k)||p'(\cdot|x_k))\Big)\Big|x_t, x_{t-1}\right]\right]$$

$$= \mathbb{E}_{x_{t-1}\sim p^*(\cdot|x_t)}[Q_{\text{soft}}^*(x_t, x_{t-1})] - \alpha D_{KL}(p^*(\cdot|x_t)||p'(\cdot|x_t)). \tag{31}$$

By substituting the soft optimal policy to Equation (31), we can get the soft Bellman equation as follows:

$$V_{\text{soft}}^*(x_t) = \mathbb{E}_{x_{t-1}\sim p^*(\cdot|x_t)}\left[Q_{\text{soft}}^*(x_t, x_{t-1}) - \alpha \log \frac{p^*(x_{t-1}|x_t)}{p'(x_{t-1}|x_t)}\right] \tag{32}$$

$$= \mathbb{E}_{x_{t-1}\sim p^*(\cdot|x_t)}\left[Q_{\text{soft}}^*(x_t, x_{t-1}) - \alpha \log \frac{\frac{1}{Z(x_t)} p'(x_{t-1}|x_t) \exp\Big(Q_{\text{soft}}^*(x_t,x_{t-1})/\alpha\Big)}{p'(x_{t-1}|x_t)}\right] \tag{33}$$

$$= \mathbb{E}_{x_{t-1}\sim p^*(\cdot|x_t)}\big[\alpha \log Z(x_t)\big] = \alpha \log Z(x_t) \tag{34}$$

$$= \alpha \log \int p'(x_{t-1} | x_t) \exp\big(Q_{\text{soft}}^*(x_t, x_{t-1})/\alpha\big) dx_{t-1} \tag{35}$$

$$= \alpha \log \mathbb{E}_{x_{t-1}\sim p'(\cdot|x_t)}\left[\exp\Big(\frac{R(x_t, x_{t-1}) + V_{\text{soft}}^*(x_{t-1})}{\alpha}\Big)\right]. \tag{36}$$

## A.3 APPROXIMATE SOFT Q-FUNCTION USING TWEEDIE'S FORMULA

Training value network is often challenging in diffusion MDP (Hu et al., 2025). Using posterior mean estimation, $\hat{x}_0(x_t)$ obtained by Tweedie's formula (Chung et al., 2023): black

$$\hat{x}_0(x_t) = \mathbb{E}_{p'}[x_0|x_t] \simeq \frac{1}{\sqrt{\bar{\alpha}_t}}\left(x_t + \sqrt{1 - \bar{\alpha}_t}\,\epsilon'_\theta(x_t, t)\right), \tag{37}$$

we can approximate the soft optimal Q function (Li et al., 2024):

$$Q_{\text{soft}}^*(x_t, x_{t-1}) = V_{\text{soft}}^*(x_{t-1}) \tag{38}$$

$$= \alpha \log \mathbb{E}_{x_0,\ldots,x_{t-2}\sim p'(\cdot|x_{t-1})}\left[\exp\left(\frac{R(x_1, x_0)}{\alpha}\right)\Big|x_{t-1}\right] \tag{39}$$

$$\approx r(\hat{x}_0(x_{t-1})). \tag{40}$$

Then we have an approximation of the soft optimal Q function in Equation (27). Therefore, by minimizing the KL-divergence in Equation (26), we can get a surrogate soft optimal policy. The corresponding loss function is (Haarnoja et al., 2018):

$$\mathcal{L}(\theta) = \mathbb{E}_{x_t}\left[\mathbb{E}_{x_{t-1}\sim p_\theta(\cdot|x_t)}[-Q_{\text{soft}}^*(x_t, x_{t-1}) + \alpha D_{\text{KL}}(p_\theta(x_{t-1}|x_t)||p'(x_{t-1}|x_t))]\right] \tag{41}$$

$$\approx \mathbb{E}_{x_t}\left[\mathbb{E}_{x_{t-1}\sim p_\theta(\cdot|x_t)}[-r(\hat{x}_0(x_{t-1})) + \alpha D_{\text{KL}}(p_\theta(x_{t-1}|x_t)||p'(x_{t-1}|x_t))]\right]. \tag{42}$$

Then the gradient of the Equation (42) is given by:

$$\nabla_\theta \mathcal{L}(\theta) = \mathbb{E}_{x_t}\left[\mathbb{E}_{\epsilon\sim\mathcal{N}(0,I)}\left[-\nabla_{x_{t-1}}r(\hat{x}_0(x_{t-1})) \cdot \nabla_\theta \mu_\theta(x_t, t) + \alpha \nabla_\theta D_{KL}(p_\theta||p')\right]\right], \tag{43}$$

with the reparameterization $x_{t-1} = \mu_\theta(x_t, t) + \sigma_t \epsilon$

# B KL-REGULARIZED RL IN DISCOUNTED MDP

Considering the inherent stochasticity of the diffusion MDP, we introduce a discount factor $\gamma \in [0, 1)$ to improve credit assignment. Then we can rewrite the original KL-regularized RL objective in Equation (18) as follows:

$$p^* = \arg\max_{p_\theta} \mathbb{E}_{\tau \sim p_\theta(\tau)} \left[ \sum_{t=1}^{T} \gamma^{T-t} \Big( R(x_t, x_{t-1}) - \alpha \mathcal{D}_{KL}(p_\theta(\cdot \mid x_t) || p'(\cdot \mid x_t)) \Big) \right] \tag{44}$$

$$= \arg\max_{p_\theta} \mathbb{E}_{\tau \sim p_\theta(\tau)} \left[ \gamma^{T-1} r(x_0) - \sum_{t=1}^{T} \gamma^{T-t} \alpha \mathcal{D}_{KL}(p_\theta(\cdot \mid x_t) || p'(\cdot \mid x_t)) \right]. \tag{45}$$

The discount factor $\gamma$ causes the signal from the terminal reward $r(x_0)$ to decay exponentially as $t$ increases, downweighting the contribution of early-stage actions. Following the definition in Equation (22), we define the soft optimal Q-function for $(x_t, x_{t-1})$, considering the discounted return:

$$Q_{\text{soft}}^*(x_t, x_{t-1}) = r(x_t, x_{t-1}) + \mathbb{E}_{p^*} \left[ \sum_{k=1}^{t-1} \gamma^{t-k} \Big( R(x_k, x_{k-1}) - \alpha D_{KL}(p^*(\cdot|x_k)||p'(\cdot|x_k)) \Big) \Big| x_t, x_{t-1} \right] \tag{46}$$

$$= R(x_t, x_{t-1}) + \mathbb{E}_{p^*} \left[ \gamma^{t-1} r(x_0) - \sum_{k=1}^{t-1} \gamma^{t-k} \alpha D_{KL}(p^*(\cdot|x_k)||p'(\cdot|x_k))) \Big| x_t, x_{t-1} \right]. \tag{47}$$

The derivation in (21 - 23) simplifies the global objective into a per-step maximization problem by using the soft optimal Q-function. This derivation still holds for our discounted MDP, yielding the same soft optimal policy $p^*(\cdot \mid x_t)$:

$$p^*(\cdot \mid x_t) = \frac{p'(\cdot \mid x_t) \exp\big(Q_{\text{soft}}^*(x_t, \cdot)/\alpha\big)}{\int p'(x_{t-1} \mid x_t) \exp\big(Q_{\text{soft}}^*(x_t, x_{t-1})/\alpha\big) \, dx_{t-1}}. \tag{48}$$

## B.1 SOFT BELLMAN EQUATION IN DISCOUNTED MDP

Again, we can define a soft value function with discounted return as follows:

$$V_{\text{soft}}^*(x_t) = \mathbb{E}_{p^*} \left[ \sum_{k=1}^{t} \gamma^{t-k} \Big( R(x_k, x_{k-1}) - \alpha \, D_{\text{KL}}\big(p^*(\cdot \mid x_k) \,\big\|\, p'(\cdot \mid x_k)\big) \Big) \Big| x_t \right] \tag{49}$$

$$= \mathbb{E}_{p^*} \left[ \gamma^{t-1} r(x_0) - \sum_{k=1}^{t} \gamma^{t-k} \, \alpha \, D_{\text{KL}}\big(p^*(\cdot \mid x_k) \,\big\|\, p'(\cdot \mid x_k)\big) \Big| x_t \right] \tag{50}$$

Then we can derive the soft optimal Q-function in terms of the soft optimal value function:

$$Q_{\text{soft}}^*(x_t, x_{t-1}) = R(x_k, x_{k-1}) + \mathbb{E}_{p^*} \left[ \sum_{k=1}^{t-1} \gamma^{t-k} \Big( R(x_k, x_{k-1}) - \alpha D_{KL}(p^*(\cdot|x_k)||p'(\cdot|x_k)) \Big) \Big| x_t, x_{t-1} \right] \tag{51}$$

$$= R(x_k, x_{k-1}) + \gamma \, \mathbb{E}_{p^*} \left[ \sum_{k=1}^{t-1} \gamma^{(t-1)-k} \Big( R(x_k, x_{k-1}) - \alpha D_{KL}(p^*(\cdot|x_k)||p'(\cdot|x_k)) \Big) \Big| x_t, x_{t-1} \right] \tag{52}$$

$$= R(x_k, x_{k-1}) + \gamma \, V_{\text{soft}}^*(x_{t-1}). \tag{53}$$

Following the derivation (31 - 36), we can get the soft Bellman equation as follows:

$$V_{\text{soft}}^*(x_t) = \mathbb{E}_{\tau \sim p^*}\left[\gamma^{t-1}r(x_0) - \sum_{k=1}^{t}\gamma^{t-k}\alpha\, D_{\text{KL}}\big(p^*(\cdot \mid x_k) \,\big\|\, p'(\cdot \mid x_k)\big) \,\Big|\, x_t\right] \tag{54}$$

$$= \mathbb{E}_{x_{t-1}\sim p^*(\cdot|x_t)}\left[\mathbb{E}_{\tau \sim p^*}\left[\gamma^{t-1}r(x_0) - \sum_{k=1}^{t}\gamma^{t-k}\alpha\, D_{\text{KL}}\big(p^*(\cdot \mid x_k) \,\big\|\, p'(\cdot \mid x_k)\big) \,\Big|\, x_t, x_{t-1}\right]\right] \tag{55}$$

$$= \mathbb{E}_{x_{t-1}\sim p^*(\cdot|x_t)}[Q_{\text{soft}}^*(x_t, x_{t-1})] - \alpha\, D_{\text{KL}}\big(p^*(\cdot \mid x_t) \,\big\|\, p'(\cdot \mid x_t)\big) \tag{56}$$

$$= \alpha \log \mathbb{E}_{x_{t-1}\sim p'(\cdot|x_t)}\left[\exp\big(\frac{R(x_k, x_{k-1}) + \gamma V_{\text{soft}}^*(x_{t-1})}{\alpha}\big)\right] \tag{57}$$

## B.2 BOUNDS OF SOFT Q-FUNCTION

In this section, we present the detailed derivation for the bounds of the soft optimal Q function:

$$\alpha \log \mathbb{E}_{x_{0:t-2}\sim p'(\cdot|x_{t-1})}\left[\exp\big(\frac{\gamma^{t-1}}{\alpha}r(x_0)\big)\right] \le Q_{\text{soft}}^*(x_t, x_{t-1}) \le \gamma^{t-1}\alpha \log \mathbb{E}_{x_{0:t-2}\sim p'(\cdot|x_{t-1})}\left[\exp\big(\frac{r(x_0)}{\alpha}\big)\right].$$

We can rewrite the soft Bellman equation in Equation (57) as:

$$\exp\Big(\frac{V_{\text{soft}}^*(x_t)}{\alpha}\Big) = \mathbb{E}_{x_{t-1}\sim p'(\cdot|x_t)}\left[\exp\Big(\frac{\gamma V_{\text{soft}}^*(x_{t-1})}{\alpha}\Big)\right]. \tag{58}$$

The discount factor $\gamma \in [0, 1)$ makes the function $u \to u^\gamma$ concave. Consequently, each recursive expansion of the soft Bellman equation invokes Jensen's inequality, creating a gap that yields a bound on the soft optimal Q-function, unlike the direct expansion in Section A.3. To derive the bounds, we first denote $Z_t(x_t) = \exp(V_{\text{soft}}^*(x_t)/\alpha)$. Then we can rewrite the Equation (58), and it's one step expansion as follows:

$$Z_t(x_t) = \mathbb{E}_{x_{t-1}\sim p'(\cdot|x_t)}\big[\, Z_{t-1}(x_{t-1})^\gamma\,\big], \tag{59}$$

$$Z_{t-1}(x_{t-1}) = \mathbb{E}_{x_{t-2}\sim p'(\cdot|x_{t-1})}\big[\, Z_{t-2}(x_{t-2})^\gamma\,\big]. \tag{60}$$

By Jensen's inequality:

$$Z_{t-1}(x_{t-1})^\gamma = \big(\mathbb{E}_{x_{t-2}\sim p'(\cdot|x_{t-1})}[Z_{t-2}(x_{t-2})^\gamma]\big)^\gamma \ge \mathbb{E}_{x_{t-2}\sim p'(\cdot|x_{t-1})}\Big[Z_{t-2}(x_{t-2})^{\gamma^2}\Big]. \tag{61}$$

By applying Equation (61) to Equation (59):

$$Z_t(x_t) = \mathbb{E}_{x_{t-1}\sim p'(\cdot|x_t)}\big[\, Z_{t-1}(x_{t-1})^\gamma\,\big] \tag{62}$$

$$\ge \mathbb{E}_{x_{t-1}\sim p'(\cdot|x_t)}\mathbb{E}_{x_{t-2}\sim p'(\cdot|x_{t-1})}\Big[\, Z_{t-2}(x_{t-2})^{\gamma^2}\Big] \tag{63}$$

$$= \mathbb{E}_{x_{t-1}, x_{t-2}\sim p'(\cdot|x_t)}\Big[\, Z_{t-2}(x_{t-2})^{\gamma^2}\Big]. \tag{64}$$

Then we can recursively expand Jenson's inequality and get the lower bound of $Z_t(x_t)$:

$$Z_t(x_t) \ge \mathbb{E}_{x_{t-2}, x_{t-1}\sim p'(\cdot|x_t)}\Big[\, Z_{t-2}(x_{t-2})^{\gamma^2}\Big] \tag{65}$$

$$\ge \cdots \tag{66}$$

$$\ge \mathbb{E}_{x_1, \cdots, x_{t-1}\sim p'(\cdot|x_t)}\Big[\, Z_1(x_1)^{\gamma^{t-1}}\Big] \tag{67}$$

$$= \mathbb{E}_{x_0, \cdots, x_{t-1}\sim p'(\cdot|x_t)}\left[\exp\Big(\frac{\gamma^{t-1}r(x_0)}{\alpha}\Big)\right]. \tag{68}$$

Since $Q_{\text{soft}}^*(x_t, x_{t-1}) = \gamma V_{\text{soft}}^*(x_{t-1})$, we can derive the lower bound of the soft optimal Q function:

$$Q_{\text{soft}}^*(x_t, x_{t-1}) = \gamma V_{\text{soft}}^*(x_{t-1}) \tag{69}$$

$$= \gamma Z_{t-1}(x_{t-1}) \tag{70}$$

$$\ge \gamma\mathbb{E}_{x_0, \cdots, x_{t-2}\sim p'(\cdot|x_{t-1})}\left[\exp\Big(\frac{\gamma^{t-2}r(x_0)}{\alpha}\Big)\right] \tag{71}$$

Now, we can also derive the upper bound of the soft optimal of the function using the same procedure as 59 - 71. By recursively expanding Jenson's inequality in the opposite way, we can obtain the upper bound of $Z_t(x_t)$ as follows:

$$Z_t(x_t) = \mathbb{E}_{x_{t-1} \sim p'(\cdot|x_t)} [ Z_{t-1}(x_{t-1})^\gamma ] \tag{72}$$

$$\leq \left( \mathbb{E}_{x_{t-1} \sim p'(\cdot|x_t)} [ Z_{t-1}(x_{t-1}) ] \right)^\gamma \tag{73}$$

$$\leq \cdots \tag{74}$$

$$\leq \left( \mathbb{E}_{x_1,\cdots,x_{t-1} \sim p'(\cdot|x_t)} [ Z_1(x_1) ] \right)^{\gamma^{t-1}} \tag{75}$$

$$= \left( \mathbb{E}_{x_0,\cdots,x_{t-1} \sim p'(\cdot|x_t)} \left[ \exp\left(\frac{r(x_0)}{\alpha}\right) \right] \right)^{\gamma^{t-1}}. \tag{76}$$

Then the upper bound of the soft optimal Q function is defined as:

$$Q^*_{\text{soft}}(x_t, x_{t-1}) = \gamma V^*_{\text{soft}}(x_{t-1}) \tag{77}$$

$$= \gamma Z_{t-1}(x_{t-1}) \tag{78}$$

$$\leq \gamma^{t-1} \mathbb{E}_{x_{0:t-2} \sim p'(\cdot|x_{t-1})} \left[ \exp\left(\frac{r(x_0)}{\alpha}\right) \right] \tag{79}$$

We have thus derived lower and upper bounds on the soft optimal Q-function:

$$\alpha \log \mathbb{E}_{x_{0:t-2} \sim p'(\cdot|x_{t-1})} \left[ \exp\left(\frac{\gamma^{t-1}}{\alpha} r(x_0)\right) \right] \leq Q^*_{\text{soft}}(x_t, x_{t-1}) \leq \gamma^{t-1} \alpha \log \mathbb{E}_{x_{0:t-2} \sim p'(\cdot|x_{t-1})} \left[ \exp\left(\frac{r(x_0)}{\alpha}\right) \right]. \tag{80}$$

## B.3 Approximation of Soft Optimal Q function in discounted MDP

We observe that applying Tweedie's formula to both the lower and upper bounds of the soft Q-function yields the identical first-order approximation:

$$\gamma \, \alpha \log \mathbb{E}_{x_0,\cdots,x_{t-2} \sim p'(\cdot|x_{t-1})} \left[ \exp\left(\frac{\gamma^{t-2} r(x_0)}{\alpha}\right) \right] \approx \gamma^{t-1} r(\hat{x}_0(x_{t-1})), \tag{81}$$

$$\gamma^{t-1} \, \alpha \log \mathbb{E}_{x_{0:t-2} \sim p'(\cdot|x_{t-1})} \left[ \exp\left(\frac{r(x_0)}{\alpha}\right) \right] \approx \gamma^{t-1} r(\hat{x}_0(x_{t-1})). \tag{82}$$

Therefore, we can approximate our $Q^*_{\text{soft}}(x_t, x_{t-1})$ as follows:

$$Q^*_{\text{soft}}(x_t, x_{t-1}) \approx \gamma^{t-1} r(\hat{x}_0(x_{t-1})). \tag{83}$$

## C    VISUALIZATION OF REWARD OVER-OPTIMIZATION PHENOMENA

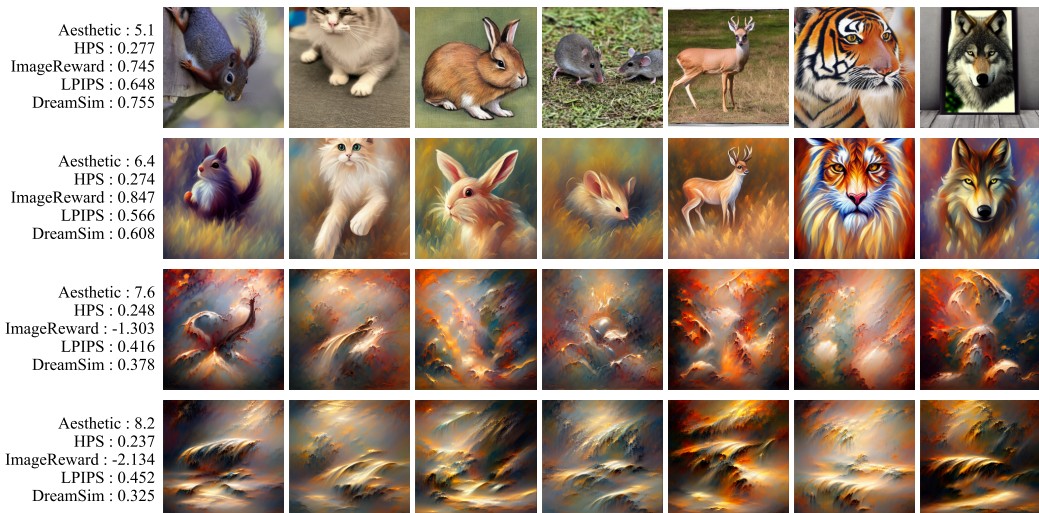

Figure 8: Visualization of reward over-optimization during fine-tuning. Each row displays generated images from four sequential training checkpoints, arranged chronologically from top to bottom. As aesthetic scores increase throughout training, the images exhibit two critical failure modes: semantic collapse (loss of alignment with prompts) and diversity collapse (convergence toward similar abstract patterns). Quantitative metrics evaluated at each checkpoint are shown on the left.

In Figure 8 we illustrate two characteristic phenomena of reward over-optimization: (1) semantic collapse and (2) diversity collapse. The examples are generated from a model fine-tuned with the DRaFT-1 method at four different training checkpoints, while the corresponding training curves are shown in Figure 3.

**Semantic collapse.** As training progresses (top to bottom), images with higher aesthetic scores progressively lose alignment with their original prompts, dissolving into abstract textures. This phenomenon is reflected in the decline of alignment metrics such as HPS and ImageReward.

**Diversity collapse.** At the same time, images with higher aesthetic scores lose diversity, with backgrounds and textures converging toward highly similar patterns. This phenomenon is reflected in the decline of diversity metrics such as LPIPS and DreamSim.

# D    EXPERIMENT DETAILS

## D.1    IMPLEMENTATION DETAILS

For all experiments, we use Stable Diffusion v1.5 (Rombach et al., 2022) as the base model and adopt LoRA (Hu et al., 2022) for compute-efficient fine-tuning. For the consistency model, we employ LCM-LoRA (Luo et al., 2023) on Stable Diffusion v1.5. We set the diffusion sampling steps to 50. All experiments are conducted on two NVIDIA RTX 4090 24 GB GPUs for text-to-image fine-tuning and on two NVIDIA A6000 48 GB GPUs for online black-box optimization. For SQDF, fine-tuning takes approximately 62 s and 401 s per update step when using aesthetic and HPS as reward functions, respectively, and 490 s per inner update step for online black-box optimization.

### D.1.1    FINETUNING FOR TEXT-TO-IMAGE DIFFUSION MODELS

For all baselines on text-to-image tasks, we use the AdamW optimizer (Loshchilov & Hutter, 2019) parameters with $\beta_1 = 0.9$, $\beta_2 = 0.999$, weight decay of 0.0001, and set the classifier-free guidance (CFG) scale to 5.0 for all methods. Following (Clark et al., 2023), we divide our experiments into two distinct settings. For our large-scale experiments, which involve fine-tuning on the HPS dataset, we used a batch size of 258, a LoRA rank of 32, and 500 training steps. For the small-scale experiments, the aesthetic score, we used a batch size of 64, a LoRA rank of 4, and 2000 training steps, except for DDPO, where we use 500 training steps, which takes the same wall time as SQDF. We provide implementation details for all experimental methods below. To ensure fair comparison and reproducible results, we used official codebases and author-recommended hyperparameters whenever available.

**SQDF**    For our implementation of SQDF, we utilized DDPM sampling with 50 steps. In small-scale experiments, we set the discount factor to $\gamma = 0.9$, the KL-regularization coefficient to $\alpha = 2$, and the learning rate to $1e - 3$. In large-scale experiments, these values were adjusted to $\gamma = 0.93$ and $\alpha = 0.05$, and the learning rate was set to $5e - 4$, respectively.

**DRaFT**    As there is no official implementation available for DRaFT, we use the AlignProp codebase, a methodologically similar concurrent work. We utilized DDIM sampling with 50 steps. The original paper notes that there is no significant performance difference between DDIM and DDPM samplers. Following the settings outlined in their paper, we used a learning rate of $4e - 4$ for small-scale experiments and $2e - 4$ for large-scale experiments.

**ReFL**    We implemented ReFL on top of the AlignProp codebase. We used DDPM sampling with 50 steps. The original ReFL method generates samples by denoising to a random timestep $t$ and then performing a one-step prediction. To adapt this to our 50-step diffusion process, we set $t$ by sampling it uniformly from the range $[40, 50]$. For the small-scale setting, we used a learning rate of $1e - 5$. As the original paper does not provide hyperparameters for a large-scale setting, and we observed slow convergence with the smaller learning rate, we used $2e - 4$ to match the DRaFT setup.

**DDPO**    For DDPO, we utilized the official codebase provided by the authors. We employed DDPM sampling with 50 steps. We used a learning rate of $3e - 4$ for both small-scale and large-scale experiments. This value was chosen as it is the recommended setting for reproducing results in the official GitHub repository. Since no specific hyperparameter was provided for large-scale experiments, we extended this recommended setting to that context as well.

**KL-regularized baselines**    For DRaFT+KL and DDPO+KL, we introduce KL-regularization into the original objective same as our method:

$$p^* = \arg\max_{p_\theta} \mathbb{E}_{\tau \sim p_\theta(\tau)} \left[ r(x_0) - \alpha \sum_{t=1}^{T} \mathcal{D}_{KL}(p_\theta(\cdot \mid x_t) || p'(\cdot \mid x_t)) \right],$$

Meanwhile, the learning method differed between the two methods. For DRaFT+KL, we calculated the KL divergence at each denoising step and added this value to the reward signal. We set the

default regularization coefficient to $\alpha = 0.04$, a value chosen to match the expected value of the KL term in SQDF. All other hyperparameters for DRaFT+KL were kept the same as the original DRaFT.

For DDPO+KL, we implement KL regularization by computing the KL divergence at each denoising timestep, detaching it from the computational graph, and subtracting it from the advantage before policy updates. We use a regularization coefficient of $\alpha = 0.2$, with all other parameters following the original DDPO configuration.

### D.1.2 ONLINE BLACK-BOX OPTIMIZATION WITH DIFFUSION MODELS

For fine-tuning the diffusion model, we used a learning rate of $1e - 3$ with the AdamW optimizer (Loshchilov & Hutter, 2019) of $\beta_1 = 0.9$, $\beta_2 = 0.999$, and a weight decay of 0.1, and set the classifier-free guidance scale to 7.5. For training the reward proxy, we used an MLP on top of a frozen CLIP encoder (Radford et al., 2021). To incorporate an uncertainty bonus, we implemented two techniques, (1) Bootstrap: statistical ensemble of neural networks, created by resampling the given dataset, and (2) a UCB bonus derived from the final neural network layer. We adopt the number of bootstrap head = 4, and UCB parameter $C_1 = 0.01$, $\lambda = 0.001$.

**SQDF** For our implementation of SQDF, we set a discount factor to $\gamma = 0.9$ and a KL-regularization coefficient to $\alpha = 1$. For each outer loop, we set inner iteration=20 and samples per iteration = 64. We adopt a prioritized replay buffer for training, as detailed in Section E.1.

**SEIKO** We adopted the hyperparameter settings from the official codebase for SEIKO (Uehara et al., 2024c). Specifically, we set the KL-regularization coefficient to $\alpha = 0.01$ and performed 5 inner iterations with 64 samples per each. The truncated back-propagation step K was uniformly sampled from (0,50), where K limits the number of steps through which gradients are backpropagated.

**PPO+KL** We implement KL-penalized PPO (Schulman et al., 2017a) as mentioned in Section D.1.1 and train it directly using the oracle reward with the same budget. We employ the hyperparameter settings from the official DDPO codebase, adding the KL-regularization with $\alpha = 0.2$.

### D.2 TASK AND PLOTTING DETAILS

In this section, we provide a detailed explanation of the tasks, along with corresponding figures that offer further clarification for the interpretation.

### D.2.1 TOY TASK

The toy example in the second row of the Figure 2 demonstrates our intuition of employing the consistency model. We use a 2D Gaussian Mixture Model (GMM) as the ground truth data distribution. GMM contains nine components with means positioned on a regular $3 \times 3$ grid spanning from $(-4, -4)$ to $(4, 4)$, where each component has an isotropic covariance of 0.3. We visualize one-step predictions to a clean sample ($t = 0$) from various noise levels ($t = 35, 25, 15$).

### D.2.2 TEXT-TO-IMAGE FINE-TUNING

For both tasks in Figure 3, we plot six points for each method, using checkpoints taken at uniform intervals up to a defined terminal rounds. While the aesthetic score is independent of the alignment and diversity score, HPS has a positive dependency on the alignment score. To this end, we experiment and plot Figure 3 with different criteria. For the top row (optimizing for aesthetic score), the terminal epoch was set as the point at which each method first reached a target reward of 8.0, where baselines exhibit severe semantic collapse. If not achieved, we report the maximum reward. For the bottom row (optimizing for HPS), the terminal epoch was set to the maximum training epochs of 500 for all methods.

**KL-regularized baselines comparison** In Figure 4, each data point on the trade-off curves represents the final performance of a model finetuned with a specific KL-regularization coefficient $\alpha$.

Darker points correspond to larger values of $\alpha$. For SQDF, we started with the default $\alpha = 2$ and increased it to 3 and 4 to depict performance at lower target reward points. For DRaFT+KL, we started with $\alpha = 0.03$, then increased the coefficient to $0.035$ and $0.04$. For DDPO+KL, we conducted an empirical search and plotted three points using $\alpha$ values of [0.2, 0.3, 0.4]. We note that, as different baselines have different scales of parameter settings that work well, we adjust as best as possible for the other baselines.

### D.2.3 ONLINE BLACK-BOX OPTIMIZATION

Following (Uehara et al., 2024c), our online black-box optimization task employs an iterative procedure, which is repeated over 4 outer loops. We assume the Aesthetic score as a black-box oracle reward function. Each loop consists of 4 steps: (1) sampling an image from the current policy, (2) querying the oracle reward function, (3) updating the reward proxy model with oracle feedback, (4) fine-tuning the policy with the updated proxy model. We execute this process with [1024, 2048, 4096, 8192] oracle queries per loop, for a total of 15360 feedback. After the final loop, we report the optimization target reward (Aesthetic Score), unseen rewards (ImageReward, HPS), and diversity metrics in Table 1.

### D.3 TRAINING AND EVALUATION PROMPTS DETAILS

For optimizing the aesthetic score, we use the 45 simple animals dataset as training and evaluation prompts, following Black et al. (2023). For optimizing HPS, we use the HPDv2 dataset (Wu et al., 2023), which contains 3,200 prompts divided into four categories (Animation, Concept Art, Painting, and Photo), with 800 prompts per category. We split the prompts into training and evaluation prompts using a fixed random seed, sampling 12 evaluation prompts per category and using the remaining prompts for training. For online black-box optimization, we train on the 45 simple animals dataset and evaluate on another six prompts (snail, hippopotamus, cheetah, crocodile, lobster, and octopus), following the approach of (Uehara et al., 2024c).

### D.4 EVALUATION DETAILS

For evaluation, we generate 32 images per prompt from the evaluation set. To measure alignment, all sampled images are scored using the ImageReward, HPSv2, and pickscore, and we report the mean score for each metric. To measure diversity, we compute pairwise distances across all generated images. Specifically, for LPIPS, we use the perceptual distances directly produced by the model and report the mean of all pairwise values. For DreamSim, we first extract feature representations for all images and then compute the average pairwise cosine distance between these features.

# E  ADDITIONAL ANALYSIS

## E.1  OPTIMIZE WITH PRIORITIZED REPLAY BUFFER

One of the key innovations of SQDF is off-policy update with a replay buffer. In this subsection, we analyze the effect of using a prioritized replay buffer (Mnih et al., 2015; Schaul et al., 2015) on overall performance.

We prioritize samples that are from high-reward trajectories and are closer to the clean sample. Specifically, for each sample $x_t$ in the buffer, we assigned a value of $\gamma^t r$, where $r$ is the reward obtained from the trajectory's final state ($x_0$) and $t$ is the timestep. Training samples are then drawn from the buffer according to probabilities derived from these priority values.

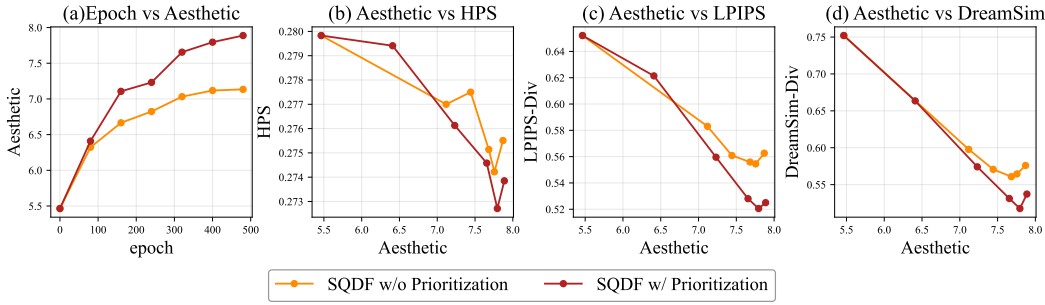

Figure 9: Effect of Prioritized buffer on Aesthetic Score.

Figure 9 shows the training dynamics of SQDF with and without a prioritized buffer when optimizing for the aesthetic score. For plots (b)-(d), we use checkpoints taken at uniform intervals up to the epoch where each method reached an aesthetic score of 8.0.

The results show that SQDF with Prioritization converges to high aesthetic scores more rapidly, but exhibits a trade-off by achieving lower alignment and diversity scores in the high-reward range (aesthetic score $> 7.0$). This analysis indicates that a prioritized buffer improves sample efficiency and suggests that integrating more advanced prioritization techniques is a promising direction for further performance enhancements.

## E.2  EFFECT OF VARYING DISCOUNT FACTOR

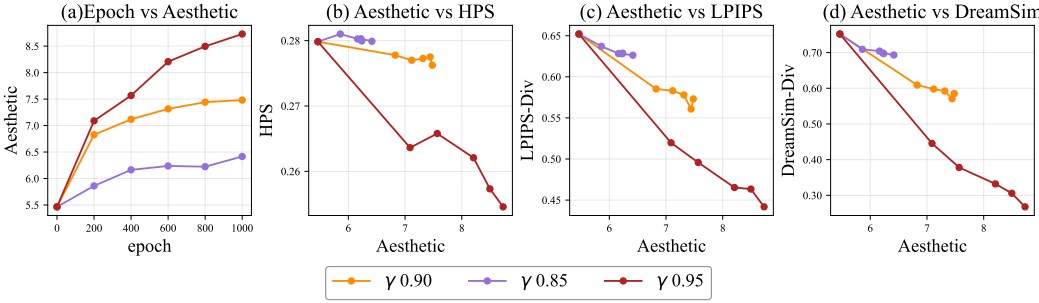

Figure 10: Training curves with different $\gamma$. A higher discount factor $\gamma$ leads to faster convergence but with less diversity and alignment score.

We investigate the effect of varying the discount factor $\gamma$ and provide a recommendation for its selection. We conduct experiments with $\gamma \in \{0.85, 0.9, 0.95\}$ under a diffusion denoising step setting of $T = 50$, using the aesthetic score as the target reward.

As shown in Figure 10, larger values of $\gamma$ lead to faster convergence and higher optimization of the target reward, but at the cost of reduced alignment and diversity scores. Conversely, smaller values

of $\gamma$ result in slower convergence, but the less aggressive reward optimization allows the model to achieve better diversity and alignment. These results suggest that the value of $\gamma$ controls a clear trade-off between optimization speed and sample quality.

### E.3 QUALITATIVE COMPARISON ON ONLINE BLACK-BOX OPTIMIZATION

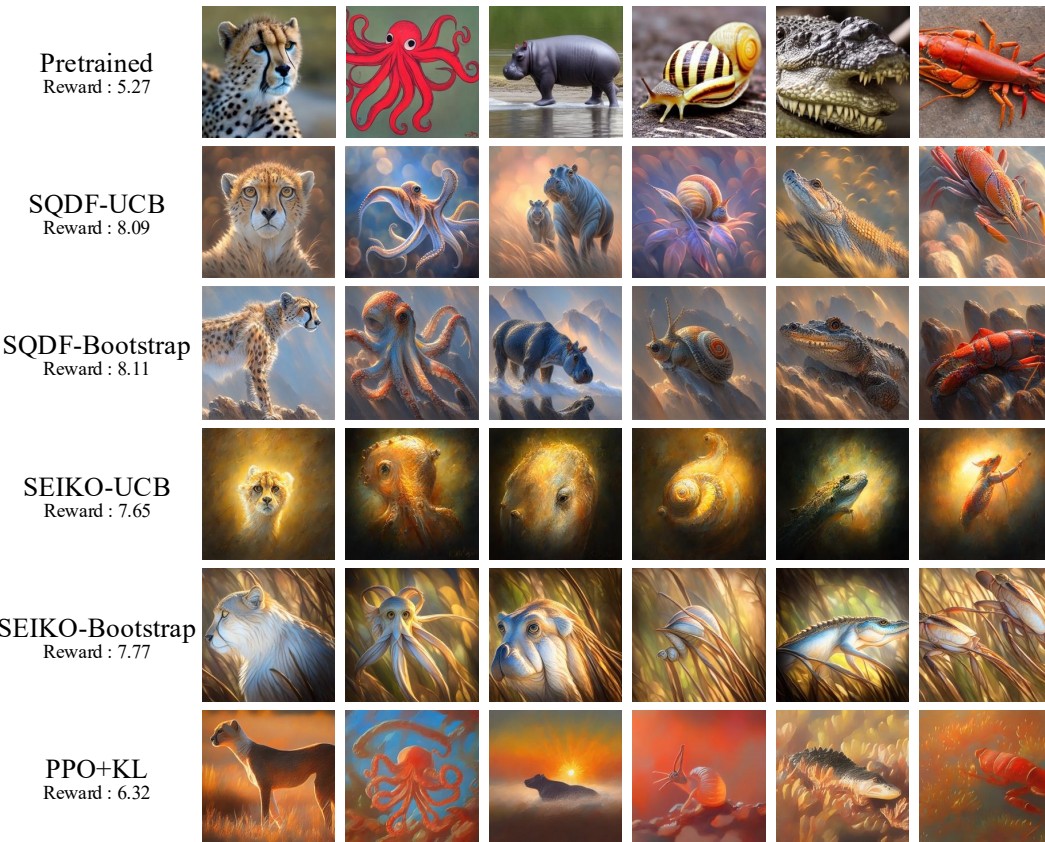

Figure 11: Qualitative comparison of online black-box optimization. Samples generated by models fine-tuned with different methods using the same oracle query budgets. Prompts: ["Cheetah", "Octopus", "Hippopotamus", "snail", "Crocodile", "Lobster"].

## E.4 QUALITATIVE COMPARISON ON HPSv2

Qualitative comparisons are presented in Figure 12, where HPS is used as the target reward. Although SQDF was optimized for a slightly lower target reward compared to other methods, it consistently produced samples that are well aligned with the input prompts. For comparison, ReFL achieved a higher target reward but showed relatively less prompt consistency. These results indicate that SQDF can effectively fine-tune under the HPS reward.

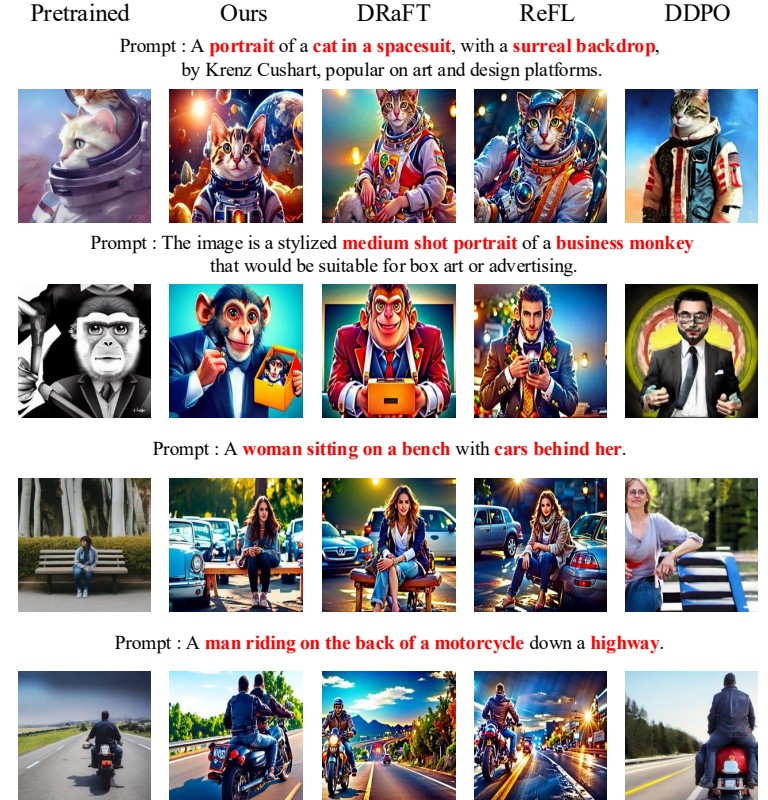

Figure 12: Qualitative comparison of different finetuning methods on the HPS.

## E.5 QUALITATIVE COMPARISON OF WITHIN-PROMPT SAMPLES

In this subsection, we visually evaluate the diversity of generated samples within individual prompts. As shown in the Figure 13 - 17, SQDF produces diverse images for the same prompt, demonstrating strong within-prompt variation. In contrast, other methods tend to generate visually similar outputs, indicating lower diversity. These results confirm that SQDF effectively preserves sample diversity and naturalness while optimizing for high target rewards.

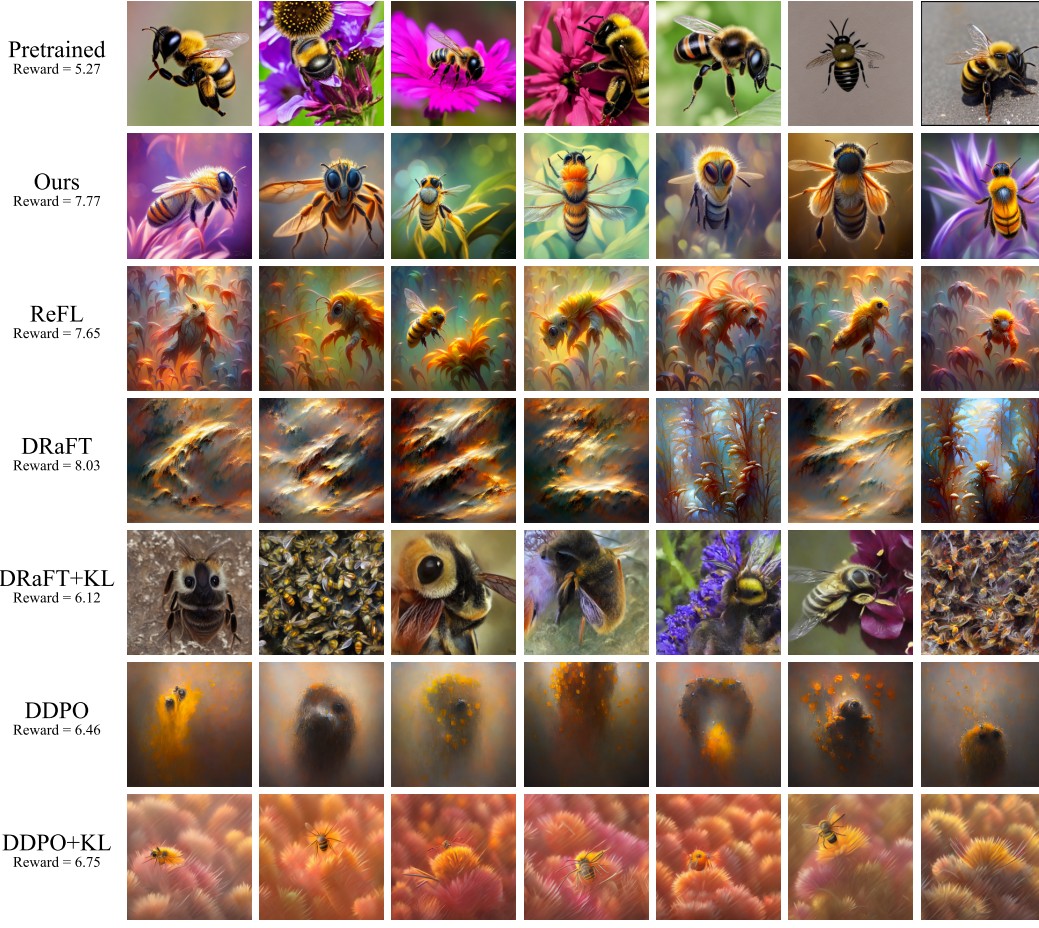

Figure 13: Samples generated by models fine-tuned with different methods for the prompt: "Bee".

Pretrained
Reward = 5.05

Ours
Reward = 8.16

ReFL
Reward = 8.17

DRaFT
Reward = 7.97

DRaFT+KL
Reward = 6.89

DDPO
Reward = 7.24

DDPO+KL
Reward = 6.94

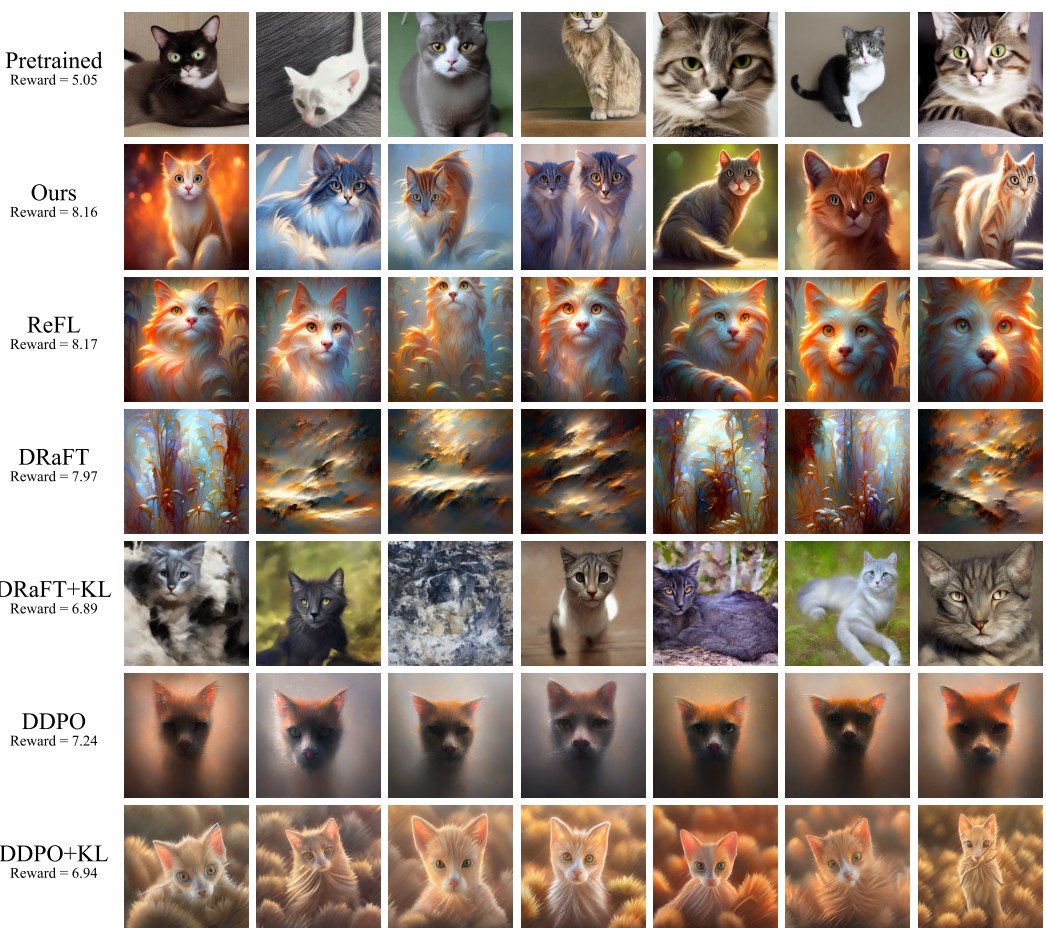

Figure 14: Samples generated by models fine-tuned with different methods for the prompt: "Cat".

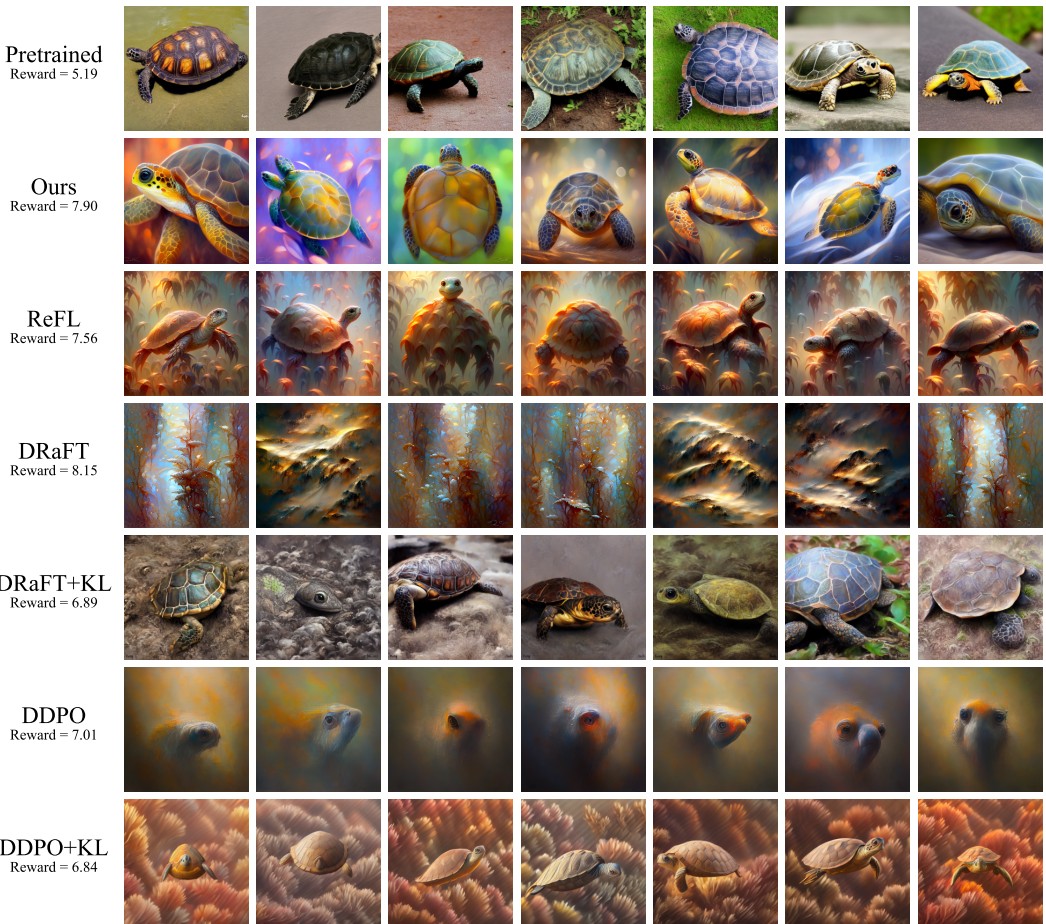

Figure 15: Samples generated by models fine-tuned with different methods for the prompt: "Turtle".

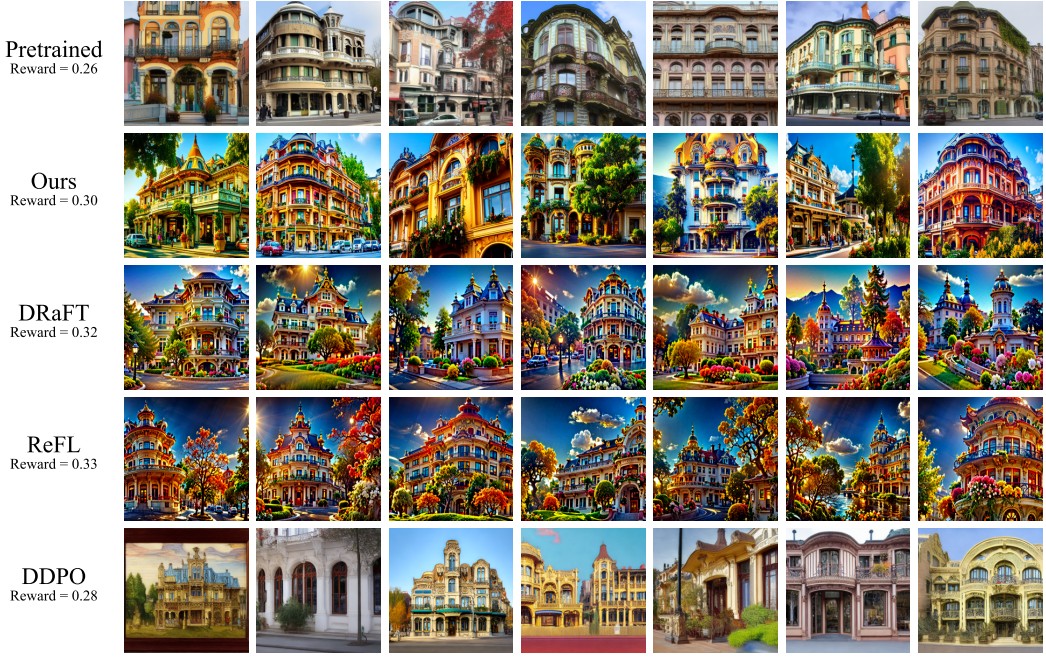

Figure 16: Samples generated by models fine-tuned with different methods for the prompt: "A landscape with an art nouveau building".

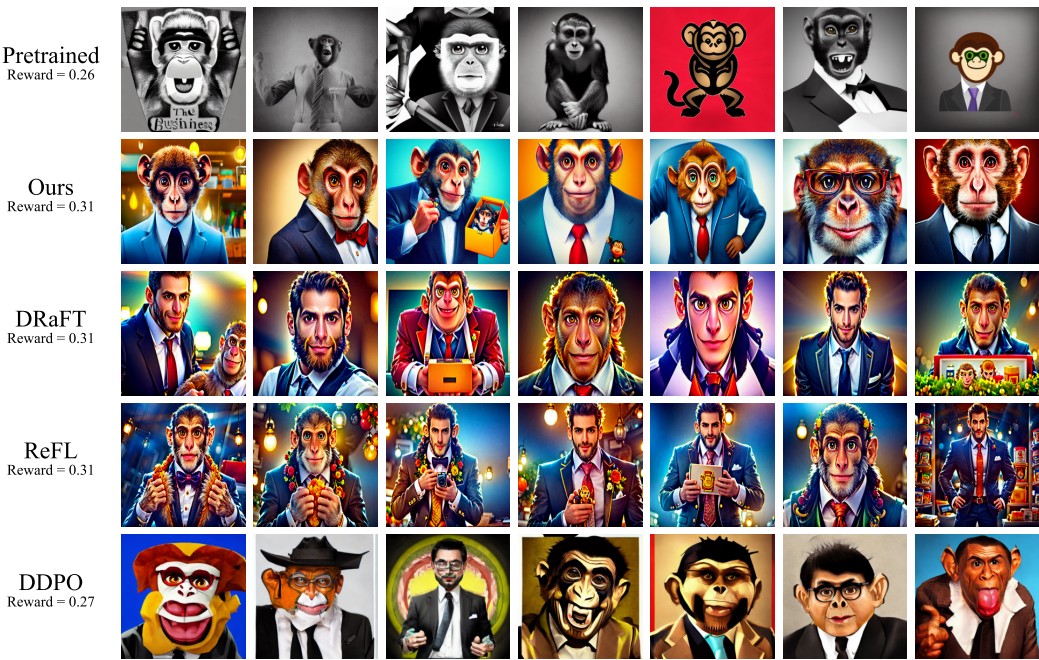

Figure 17: Samples generated by models fine-tuned with different methods for the prompt: "The image is a stylized medium shot portrait of a business monkey that would be suitable for box art or advertising".

# F EXTENSION TO ADVANCED BACKBONE

In this section, we further extend our experiments by replacing the base model from SD v1.5 to SDXL (Podell et al., 2023) to evaluate the effectiveness of SQDF on a more advanced and larger-scale architecture. All experiments are conducted using three random seeds.

First, we fine-tune SDXL using both DRaFT-1 and SQDF to optimize the aesthetic score. As shown in Table 3, SQDF outperforms DRaFT, achieving superior aesthetic scores and alignment while successfully maintaining diversity.

| Metric | SDXL (Pre-trained) | DRaFT | SQDF |
|---|---|---|---|
| Aesthetic ($\uparrow$) | 5.45 | 7.18 | **7.86** |
| ImageReward ($\uparrow$) | 0.88 | 0.91 | **1.21** |
| HPS ($\uparrow$) | 0.28 | 0.27 | **0.28** |
| LPIPS-Div ($\uparrow$) | **0.59** | 0.49 | 0.51 |
| DreamSim-Div ($\uparrow$) | **0.72** | 0.48 | 0.57 |

Table 3: Comparison of DRaFT and SQDF on fine-tuning SDXL for aesthetic score.

Furthermore, to investigate whether the effectiveness of SQDF depends on the size of the base model, we compare the relative performance improvements between SD 1.5 (860M) and SDXL (2.6B). As shown in Table 4, the relative improvement achieved by SQDF is highly consistent across both architectures.

| Metric | SD1.5 | SD1.5 + SQDF | % | SDXL | SDXL + SQDF | % |
|---|---|---|---|---|---|---|
| Aesthetic ($\uparrow$) | 5.46 | 7.87 | 44.14% | 5.45 | 7.86 | 44.22% |
| ImageReward ($\uparrow$) | 0.93 | 1.14 | 22.58% | 0.88 | 1.21 | 37.50% |
| HPS ($\uparrow$) | 0.28 | 0.28 | 0.00% | 0.28 | 0.28 | 0.00% |
| LPIPS-Div ($\uparrow$) | 0.65 | 0.56 | -13.85% | 0.59 | 0.51 | -13.56% |
| DreamSim-Div ($\uparrow$) | 0.75 | 0.58 | -22.67% | 0.72 | 0.57 | -20.83% |

Table 4: Relative performance improvements of SQDF on SD1.5 and SDXL for aesthetic score.

These results demonstrate that SQDF optimizes the target reward while mitigating over-optimization, regardless of the underlying diffusion backbone.

# G   UNCURATED SAMPLES

We present uncurated samples generated by models fine-tuned with each method discussed in the main paper.

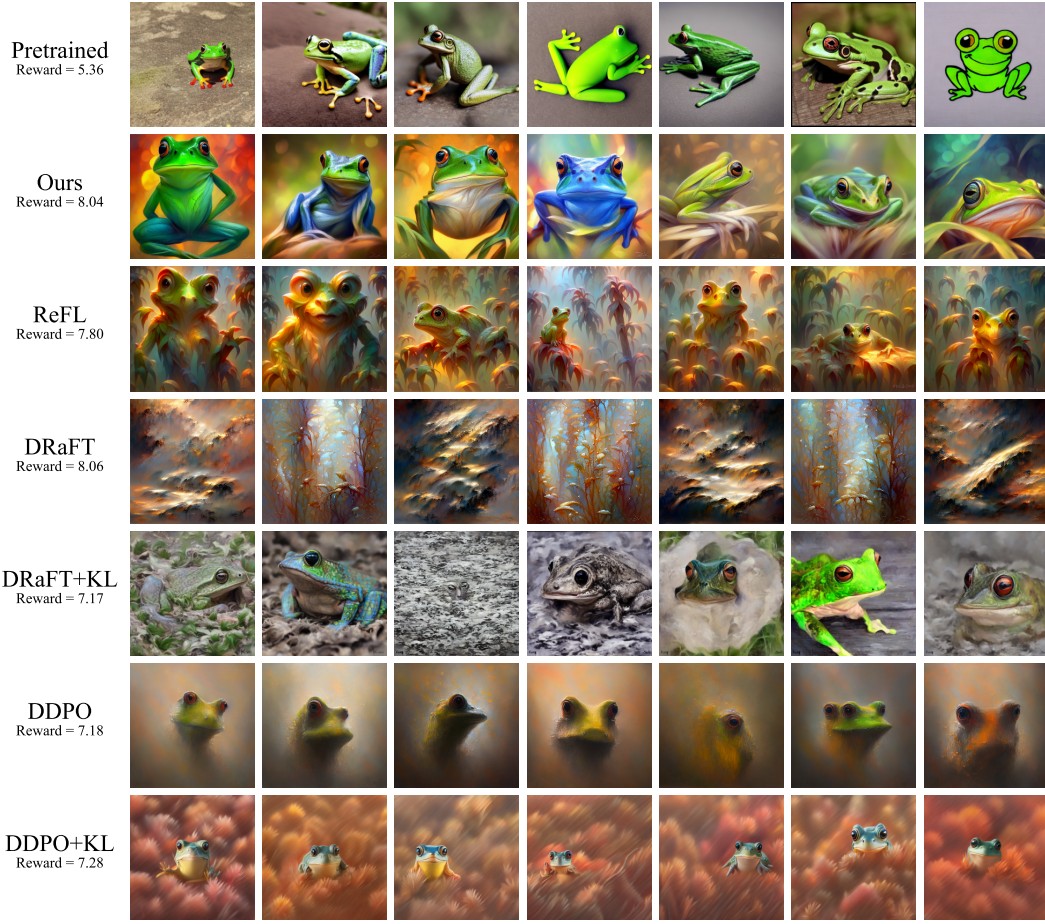

Figure 18: Uncurated samples generated by models fine-tuned with different methods for the prompt: "Frog".

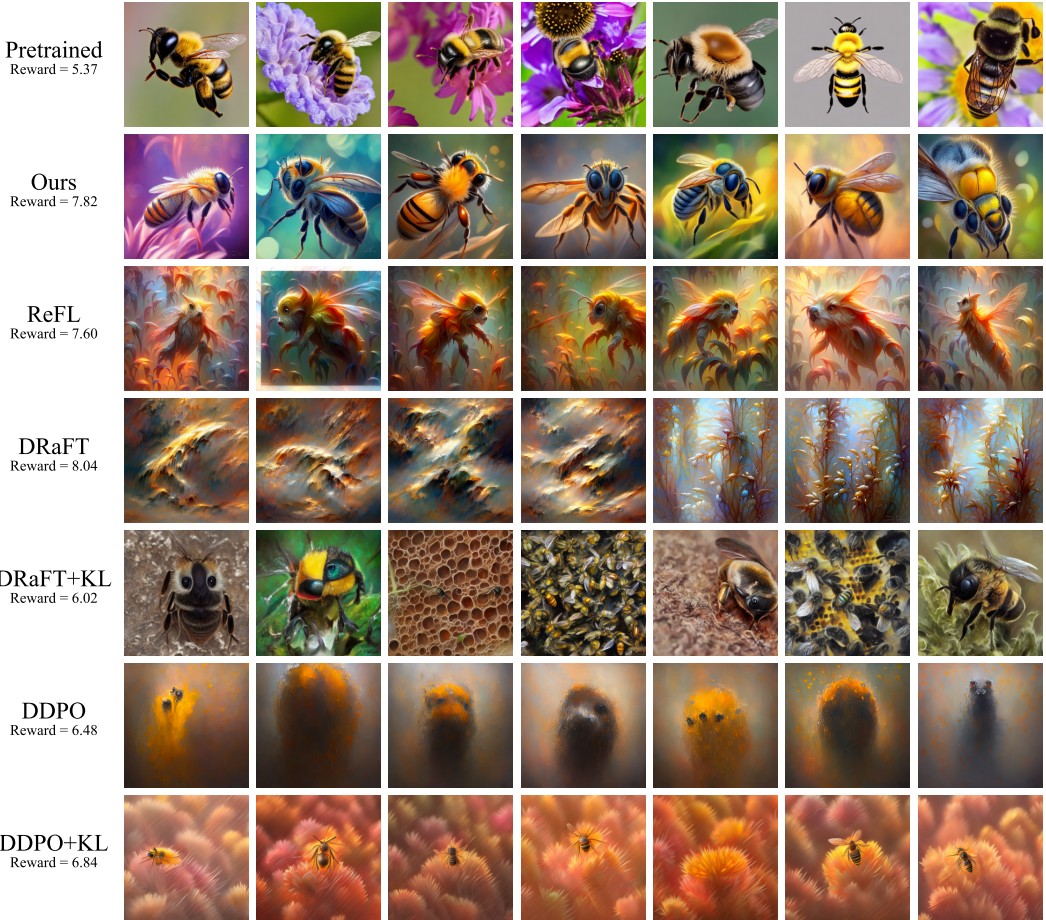

Figure 19: Uncurated samples generated by models fine-tuned with different methods for the prompt: "Bee".

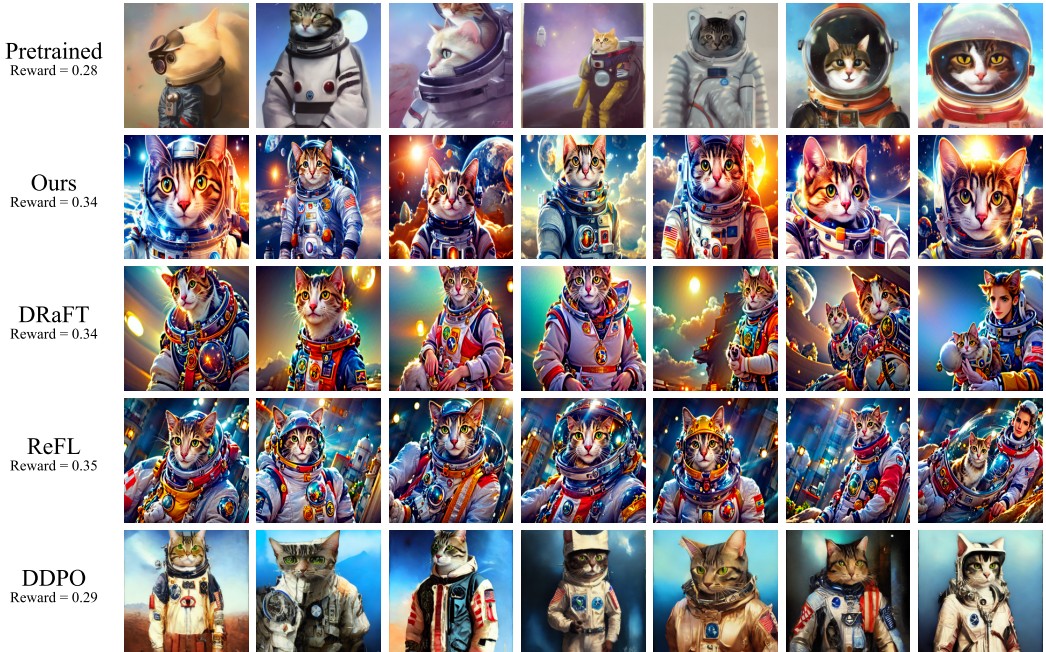

Figure 20: Uncurated samples generated by models fine-tuned with different methods for the prompt: " A portrait of a cat in a spacesuit, with a surreal backdrop, by Krenz Cushart, popular on art and design platforms.".

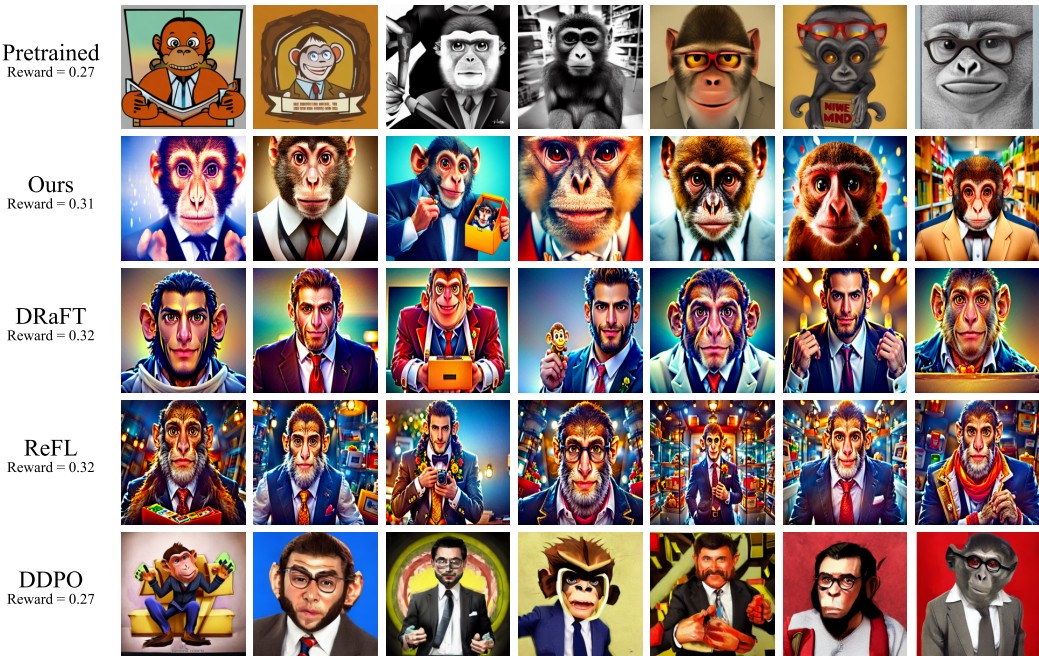

Figure 21: Uncurated samples generated by models fine-tuned with different methods for the prompt: "The image is a stylized medium shot portrait of a business monkey that would be suitable for box art or advertising".

# H   PSEUDO CODE OF SQDF

In this section, we provide pseudo codes of SQDF of the text-to-image fine-tuning task in Algorithm 1, and online black box optimization task in Algorithm 2.

---

**Algorithm 1** Soft Q Diffusion Finetuning (SQDF)

---

1: **Input:** Pretrained diffusion model $p'$; Trainable diffusion model $p_\theta$; Replay buffer $\mathcal{D}$; Training epochs $N$; Batch size $B$; Reward model $r_\phi$; Consitency model $f_\psi$;
2: Initialize $p_\theta \leftarrow \texttt{deepcopy}(p')$, replay buffer $\mathcal{D} \leftarrow \emptyset$
3: **for** $k = 1, \ldots, N$ **do**
4:     ***On-policy Sampling***
5:     **for** $b = 1, \ldots, B$ **do**
6:         Sample noise $x_T \sim \mathcal{N}(0, I)$
7:         Run diffusion denoising steps using $p_\theta$ to obtain trajectory $\{x_T, x_{T-1}, \ldots, x_1\}$
8:         For each $(x_t, t)$, store into $\mathcal{D}$
9:     **end for**
10:    *Policy Update*
11:    Sample a batch $\{(x_t^j, t^j)\}_{j=1}^B$ from $\mathcal{D}$.
12:    **for** $j = 1, \ldots, B$ **do**
13:        Perform one-step denoising $x_{t-1}^j \sim p_\theta(x_{t-1}|x_t^j, t^j)$
14:        Estimate the clean sample $\hat{x}_0^j$ via consistency model $f_\psi(x_{t-1}^j)$
15:        Compute reward $r^j = r_\phi(\hat{x}_0^j)$
16:        Compute the loss $\ell^j$ (Eq. 10)
17:    **end for**
18:    Update $\theta$ by minimizing $L = \frac{1}{B} \sum_j \ell^j$
19: **end for**

---

---

**Algorithm 2** SQDF with Online Black-Box Optimization

---

1: **Input:** Pretrained diffusion model $p'$; trainable diffusion model $p_\theta$; replay buffer $\mathcal{D}$; Consitency model $f_\psi$; outer rounds $N$; inner updates per round $M$; oracle query schedule $\{O_1, O_2, \ldots, O_N\}$; batch size $B$; surrogate reward model $\hat{r}_\phi$; oracle reward $r^*$

2: Initialize $p_\theta \leftarrow \text{deepcopy}(p')$, replay buffer $\mathcal{D} \leftarrow \emptyset$, labeled dataset $\mathcal{S} \leftarrow \emptyset$

3: Collect $O_1$ initial samples $\{x_0^i\}_{i=1}^{O_1}$ using $p_\theta$, query oracle rewards $\{y^i\}_{i=1}^{O_1}$, and store data in $\mathcal{S}$

4: Train surrogate $\hat{r}_\phi$ on $\mathcal{S} = \{(x_0^i, y^i)\}_{i=1}^{O_1}$

5: **for** $k = 1, \ldots, N$ **do**

6:     **for** $m = 1, \ldots, M$ **do**

7:         ***On-policy Sampling***

8:         **for** $b = 1, \ldots, B$ **do**

9:             Sample noise $x_T \sim \mathcal{N}(0, I)$

10:             Run diffusion denoising under $p_\theta$ to obtain trajectory $\{x_T, x_{T-1}, \ldots, x_1\}$

11:             Store each state tuple $(x_t, t)$ in $\mathcal{D}$

12:         **end for**

13:         ***Policy Update***

14:         Sample a batch $\{(x_t^j, t^j)\}_{j=1}^B$ from $\mathcal{D}$

15:         **for** $j = 1, \ldots, B$ **do**

16:             Perform one-step denoising $x_{t-1}^j \sim p_\theta(x_{t-1} \mid x_t^j, t^j)$

17:             Estimate clean sample $\hat{x}_0^j$ via consistency model $f_\psi(x_{t-1}^j)$

18:             Compute surrogate reward $r^j = \hat{r}_\phi(\hat{x}_0^j)$

19:             Compute the loss $\ell^j$ (Eq. 10)

20:         **end for**

21:         Update $\theta$ by minimizing $L = \frac{1}{B} \sum_j \ell^j$

22:     **end for**

23:     ***Oracle Querying and Surrogate Update***

24:     **for** $o = 1, \ldots, O_k$ **do**

25:         Sample noise $x_T \sim \mathcal{N}(0, I)$

26:         Run diffusion denoising under $p_\theta$ to obtain a clean sample $x_0$

27:         Query the oracle to obtain reward $y = r^*(x_0)$

28:         Store $(x_0, y)$ in $\mathcal{S}$

29:     **end for**

30:     Retrain surrogate $\hat{r}_\phi$ on the dataset $\mathcal{S}$

31: **end for**

---

