# OpenReview forum: "Diffusion Fine-Tuning via Reparameterized Policy Gradient of the Soft Q-Function"
_ICLR.cc/2026/Conference — ICLR 2026 Poster_

### Official Review · Reviewer_5VJm · 2025-10-31

**Soundness:** 3
**Presentation:** 3
**Contribution:** 2
**Rating:** 4
**Confidence:** 4

**Summary:**

This paper introduces Soft Q-learning for Diffusion Fine-tuning (SQDF), a reinforcement learning–based framework designed to fine-tune diffusion models while mitigating reward over-optimization. SQDF leverages a KL-regularized RL objective, and introduces a discount factor for better credit assignment, a consistency model for improved Q-function approximation, and an off-policy replay buffer.

**Strengths:**

* Nice Presentation: The paper addresses reward over-optimization, which is an important issue in diffusion model reinforcement learning. Moreover, the writing of this paper is very easy to understand.
* Strong Theoretical Grounding: The method is derived from diffusion model MDP formulations, showing solid theoretical consistency.
* Comprehensive Experiments: Results are thorough, including extensive qualitative and quantitative evaluations, and ablation studies.

**Weaknesses:**

* Base Model: The paper adopts Stable Diffusion 1.5, an older base model, which raises concerns about the generalizability of the proposed method to more advanced diffusion architectures.
* Clarity: Some equations and algorithm steps (e.g., Eq. 10-12, Line 14 of Algo.1) could be described more intuitively, as the current notation may make it harder for non-expert readers to follow.

**Questions:**

There are several concerns that may significantly affect the overall evaluation of this paper.
* In Section 4.1, what is the key difference between the proposed loss (Eq. 11) and the DPOK loss (Eq. 8 in [1])? It seems that the proposed loss allows the use of the reward gradient of the predicted $x_0$ at each timestep. However, the use of reward gradients has already been introduced in DRaFT [2], and the use of rewards on the predicted $x_0$ has been proposed in ReFL [3]. This makes Section 4.1 appear to be a combination of DPOK, DRaFT, and ReFL.
* In Section 4.2.1, the authors introduce a discount factor to emphasize that later timesteps are more important. However, prior work has also introduced a discount factor but emphasized the earlier denoising steps instead (Eq. 1 in [4]). Could the authors clarify this contradiction?
* In Section 4.2.2, the authors introduce a consistency model for better $x_0$ prediction。While the $x_0$ predicted by the consistency model may indeed be cleaner, it is no longer the prediction of the original model $p_\theta$. Why should the reward obtained from this predicted $x_0$ be valid for updating $p_\theta$? (Even though the images generated by the consistency model and the original model are largely similar, differences always exist in some details, and those details could be crucial for reward evaluation sometimes.)
* In Section 4.2.3, the authors introduce a replay buffer. In the conclusion, the paper claims that the replay buffer can manage the reward–diversity trade-off. However, according to Appendix E.1, the replay buffer only weights each sample by $\gamma^t r$, which seems to be equivalent to directly weighting the samples by $\gamma^t r$ in the loss function. Could the authors provide a more detailed explanation about managing the reward-diversity trade-off?

[1] DPOK: Reinforcement Learning for Fine-tuning Text-to-Image Diffusion Models.
[2] Directly Fine-Tuning Diffusion Models on Differentiable Rewards.
[3] ImageReward: Learning and Evaluating Human Preferences for Text-to-Image Generation.
[4] A Dense Reward View on Aligning Text-to-Image Diffusion with Preference.

---

> ### Author Response · Authors · 2025-11-20
>
> We appreciate the reviewer’s insightful comments and constructive feedback on our work.
>
> >**(Weakness 1)** Base Model: The paper adopts Stable Diffusion 1.5, an older base model, which raises concerns about the generalizability of the proposed method to more advanced diffusion architectures.
>
> To evaluate whether our method is effective for advanced architectures, we conduct additional experiments on SDXL [1]. First, we fine-tune SDXL with DRaFT and SQDF for the aesthetic score. As shown in the table below, SQDF outperforms the DRaFT, achieving superior aesthetic score and alignment scores while maintaining diversity.
>
> **Table A. Comparison of DRaFT and SQDF on fine-tuning SDXL for aesthetic score.**
> | Metric        | SDXL (Pre-trained) | DRaFT | SQDF     |
> | ------------- | ---- | ----- | -------- |
> | Aesthetic (↑)   | 5.45 | 7.04  | **7.81** |
> | ImageReward (↑) | 0.88 | 0.98  | **1.03** |
> | HPS (↑)         | 0.28 | 0.27  | **0.28** |
> | LPIPS-Div (↑)   | **0.59** | 0.50  | 0.50 |
> | DreamSim-Div (↑)    | **0.72** | 0.48  | 0.56 |
>
>
> Furthermore, we investigate whether the effectiveness of SQDF depends on the size of the backbone models by comparing the relative performance improvements between the SD1.5 (860M) and SDXL (2.6B). Importantly, the relative improvement achieved by SQDF is highly consistent across both architectures.
>
> **Table B. Relative performance improvements of SQDF on SD1.5 and SDXL for aesthetic score.**
>
> | Metric          | SD1.5 | SD1.5 + SQDF | % | SDXL | SDXL + SQDF | % |
> |-----------------|--------|----------------|-----|--------|-----------------|------|
> | Aesthetic (↑)  |  5.46  |   7.87  | 44.14%    |  5.45 |   7.81 |   43.30%   |
> | ImageReward (↑) | 0.93|  1.14  |   22.58%  |  0.88   |  1.03  |17.05%  |
> | HPS (↑)  |  0.28      |  0.28   |   0.00%   |    0.28    |  0.28  |   0.00%   |
> | LPIPS-Div (↑)  |  0.65      |0.56  | -13.85%  |0.59 |  0.50 |-15.25%    |
> | DreamSim-Div (↑)  |   0.75     |  0.58 |  -22.67%  |   0.72 | 0.56|-22.22% |
>
> These results demonstrate that SQDF optimizes the target reward while mitigating over-optimization, regardless of the underlying diffusion backbone.
>
> [1] Dustin Podell, Zion English, Kyle Lacey, Andreas Blattmann, Tim Dockhorn, Jonas M¨uller, Joe Penna, and Robin Rombach. Sdxl: Improving latent diffusion models for high-resolution image synthesis. In The Twelfth International Conference on Learning Representations, 2024.
>
> >**(Weakness 2)** Clarity: Some equations and algorithm steps (e.g., Eq. 10-12, Line 14 of Algo.1) could be described more intuitively, as the current notation may make it harder for non-expert readers to follow.
>
>
> Thank you for your suggestion regarding clarifications. To improve clarity, we add more detail explanation about reparameterization policy gradient above the Eq.12 in Section 4.1. We also corrected minor errors in Algorithm 1 (line 1, 14) & 2 (line 1, 17), clarifying that SQDF employs the consistency model rather than Tweedie’s formula. We appreciate your careful observations, and please let us know if you have any additional suggestions for improving clarity.

---

> ### Author Response · Authors · 2025-11-20
>
> >**(Question 1-1)**  In Section 4.1, what is the key difference between the proposed loss (Eq. 11) and the DPOK loss (Eq. 8 in [1])?
>
> While both DPOK and SQDF are built upon a KL-regularized RL objective, they diverge fundamentally in how they estimate the gradient of the return. DPOK treats the reward as a black-box function. Consequently, it must resort to REINFORCE-style gradient estimators like PPO [2][3], which rely on the likelihood ratio trick. This approach is well-known to suffer from high variance, leading to unstable updates and poor sample efficiency [4]. In contrast, SQDF leverages the fact that many practical reward functions such as aesthetic scores [5] or human preference models [6][7], which are differentiable. This allows SQDF to directly use the gradient of the reward function as the gradient of the soft Q-function via reparameterization, avoiding high-variance estimators like PPO and enabling significantly more sample-efficient and stable optimization.
>
> >**(Question 1-2)**  ... the use of reward gradients has already been introduced in DRaFT [8], and the use of rewards on the predicted $x_0$has been proposed in ReFL [7]. This makes Section 4.1 appear to be a combination of DPOK, DRaFT, and ReFL.
>
> As you noticed, DRaFT [8] and ReFL [7] already use the reward gradient for optimizing the diffusion model, and the optimization mechanism of SQDF resembles ReFL in terms of using the $\hat x_0$ for receiving the reward signal. However, one of our contributions is providing the perspective that the gradient of a differentiable reward function can be interpreted as the gradient of a training-free soft Q-function under the KL-regularized RL formulation. This perspective is absent in prior works. Furthermore, our results in Figure 4 show that DRaFT+KL is Pareto dominated by SQDF, demonstrating that SQDF is a more effective and robust optimization method than the combination of direct backpropagation and KL-regularization.
>
>
>
> [1] Fan, Ying, et al. "Dpok: Reinforcement learning for fine-tuning text-to-image diffusion models." Advances in Neural Information Processing Systems 36 (2023): 79858-79885.
>
> [2] Williams, Ronald J. "Simple statistical gradient-following algorithms for connectionist reinforcement learning." Machine learning 8.3 (1992): 229-256.
>
> [3] Schulman, John, et al. "Proximal policy optimization algorithms." arXiv preprint arXiv:1707.06347 (2017).
>
> [4] Schulman, John, et al. "High-dimensional continuous control using generalized advantage estimation." arXiv preprint arXiv:1506.02438 (2015).
>
> [5] Schuhmann, Christoph, and Romain Beaumont. "Laion-aesthetics." LAION. AI (2022).
>
> [6] Wu, Xiaoshi, et al. "Human preference score v2: A solid benchmark for evaluating human preferences of text-to-image synthesis." arXiv preprint arXiv:2306.09341 (2023).
>
> [7] Xu, Jiazheng, et al. "Imagereward: Learning and evaluating human preferences for text-to-image generation." Advances in Neural Information Processing Systems 36 (2023): 15903-15935.
>
> [8] Clark, Kevin, et al. "Directly Fine-Tuning Diffusion Models on Differentiable Rewards." The Twelfth International Conference on Learning Representations.
>
>
> >**(Question 2)**  In Section 4.2.1, the authors introduce a discount factor to emphasize that later timesteps are more important. However, prior work has also introduced a discount factor but emphasized the earlier denoising steps instead (Eq. 1 in [1]). Could the authors clarify this contradiction?
>
> While [1] focuses primarily on reward maximization, our objective is to optimize reward while preserving both diversity and alignment quality. Maintaining diversity requires preserving the pre-trained model’s marginal distribution $p(x_t)$ at early timesteps, since these stages critically influence sample diversity [2][3]. Following this insight, we downweight the credit at early denoising steps. As shown in our ablation study in section 5.3, increasing the weight on early timesteps indeed boosts reward and degrades diversity, consistent with the findings of [1]. However, since our primary goal is to mitigate over-optimization rather than maximize reward at all costs, downweighting early timesteps is more suitable for our setting.
>
>
> [1] Yang, Shentao, Tianqi Chen, and Mingyuan Zhou. "A Dense Reward View on Aligning Text-to-Image Diffusion with Preference." Forty-first International Conference on Machine Learning.
>
> [2] Jena, Rohit, et al. "Elucidating optimal reward-diversity tradeoffs in text-to-image diffusion models." 2025 IEEE/CVF Winter Conference on Applications of Computer Vision (WACV). IEEE, 2025.
>
> [3] Kim, Sunwoo, Minkyu Kim, and Dongmin Park. "Test-time Alignment of Diffusion Models without Reward Over-optimization." The Thirteenth International Conference on Learning Representations.

---

> ### Author Response · Authors · 2025-11-20
>
> >**(Question 3)**  In Section 4.2.2, the authors introduce a consistency model for better $x_0$ prediction。While the $x_0$ predicted by the consistency model may indeed be cleaner, it is no longer the prediction of the original model $p_{\theta}$. Why should the reward obtained from this predicted  $x_0$ be valid for updating $p_{\theta}$? (Even though the images generated by the consistency model and the original model are largely similar, differences always exist in some details, and those details could be crucial for reward evaluation sometimes.)
>
>
> Following Theorem 1 of [1], let's assume that the consistency model $f_\psi$ satisfies the Lipschitz condition, the step size is sufficiently small, and the training loss converges to zero. Under these conditions, the consistency model asymptotically approximates the empirical Probability Flow ODE (PF ODE) learned by the diffusion loss with an approximation error of $\mathcal O(\Delta  t^p)$, where $p$ is the order of the ODE. Therefore, assuming the usage of a powerful consistency model, we can approximate the denoising chain from the pre-trained model $p'$ with the consistency model $f_\psi$ with error bounded by $\mathcal O(\Delta  t^p)$.
>
> As you pointed out, the consistency model may deviate from the PF ODE in practice. However, our experimental results suggest that the consistency model still offers a clear advantage. We empirically compare two strategies for estimating $\hat x_0$ in Figure 7: Consistency Models and multi-step DDIM denoising. Although the CM predictions may differ from multi-step DDIM, the latter is impractical for reparameterized policy gradient because it requires backpropagation through the denoising chain, making the optimization unstable.
>
> [1]  Song, Yang, et al. "Consistency Models." International Conference on Machine Learning. PMLR, 2023
>
> >**(Question 4-1)** In Section 4.2.3, the authors introduce a replay buffer. In the conclusion, the paper claims that the replay buffer can manage the reward–diversity trade-off. However, according to Appendix E.1, the replay buffer only weights each sample by $\gamma^tr$, which seems to be equivalent to directly weighting the samples by $\gamma^tr$ in the loss function.
>
> As you pointed out, directly weighting samples by $\gamma^t r$ is equivalent in expectation to applying the same weight inside the loss function. However, according to [1], multiplying the loss by $\gamma^t r$ instead of sampling from the prioritized buffer leads to higher gradient variance, which makes optimization less sample-efficient. Table C shows that SQDF with prioritized buffer outperforms directly weighting $\gamma^t r$ to the loss function.
>
> [1] Fujimoto, Scott, David Meger, and Doina Precup. "An equivalence between loss functions and non-uniform sampling in experience replay." Advances in neural information processing systems 33 (2020): 14219-14230.
>
> >**(Question 4-2)** Could the authors provide a more detailed explanation about managing the reward-diversity trade-off?
>
> As shown in Table C, a uniform replay buffer enhances the diversity by reusing the past experience at the cost of reward. Prioritizing with $\gamma^tr$ changes the sampling distribution, making the high-reward sample used more often, and improving reward optimization at the cost of diversity.
>
> **Table C. Comparison of the four replay buffer strategies.**
> | Metric          |SQDF (w/o Buffer)|SQDF (Uniform Buffer)|SQDF (directly weighted) | SQDF (Prioritized Buffer)|
> |----------------|-|--------|-------------|---------|
> | Aesthetic (↑)   |7.40|7.15|  7.50  |  **7.89**  |
> | ImageReward (↑) |1.12|**1.23**|  1.12  |  1.08  |
> | HPS (↑)         |0.28|**0.28**|  0.27  |  0.27  |
> | LPIPS-Div (↑)   |0.58|**0.60**|  0.55  |  0.54  |
> | DreamSim-Div (↑)    |0.56|**0.59**|  0.54  |  0.53  |

---

### Official Review · Reviewer_3B3B · 2025-11-01

**Soundness:** 2
**Presentation:** 2
**Contribution:** 2
**Rating:** 6
**Confidence:** 2

**Summary:**

The paper introduces Soft-Q Diffusion Finetuning (SQDF), a novel method for aligning diffusion models with downstream reward functions while mitigating over-optimization. The core innovation is a training-free approximation of the soft Q-function using a single-step posterior mean estimate, which enables direct gradient-based optimization without the instability of value network training or high-variance Monte Carlo estimators. Experiments on text-to-image tasks demonstrate that SQDF achieves superior reward-diversity trade-offs compared to existing RL-based and direct backpropagation methods.

**Strengths:**

1. The training-free soft Q-function approximation eliminates the need for unstable value network training.

2. The paper demonstrates effectiveness across multiple tasks (aesthetic scoring, human preference optimization, black-box settings) with thorough comparisons against relevant baselines and ablation studies validating each component.

3. Results show SQDF achieves better Pareto frontiers, optimizing target rewards while maintaining significantly better alignment and diversity metrics compared to methods that suffer from semantic and diversity collapse.

**Weaknesses:**

1. While using the consistency model to estimate the future value is intriguing, it also makes one wonder what about directly fine-tuning the consistency model, which could potentially achieve similar results more efficiently given its single-step generation capability. Also see Q1.

2. While methodologically sound, the paper provides little discussion of computational overhead, training time comparisons, or memory requirements.

**Questions:**

Why did you choose the two-model architecture instead of directly fine-tuning the consistency model? Given that consistency models can generate samples using multiple steps, did you explore whether similar reward-diversity trade-offs could be achieved more efficiently by fine-tuning the consistency model directly?

---

> ### Author Response · Authors · 2025-11-20
>
> >**(Weakness 1, Question 1)** Why did you choose the two-model architecture instead of directly fine-tuning the consistency model? Given that consistency models can generate samples using multiple steps, did you explore whether similar reward-diversity trade-offs could be achieved more efficiently by fine-tuning the consistency model directly?
>
> Thank you for this insightful suggestion. We agree that fine-tuning Consistency Models (CM) is a promising direction, as prior works [1][2] have demonstrated their potential to improve the efficiency of RL finetuning. Applying reparameterized policy gradient to a CM for mitigating over-optimization is therefore an intriguing research avenue. However, we focus on fine-tuning the diffusion model in this paper, so exploring CM fine-tuning is beyond the scope of our current paper.
>
> [1] Oertell, Owen, et al. "RL for consistency models: Faster reward guided text-to-image generation." arXiv preprint arXiv:2404.03673 (2024).
>
> [2] Shekhar, Shivanshu, and Tong Zhang. "ROCM: RLHF on consistency models." arXiv preprint arXiv:2503.06171 (2025).
>
> > **(Weakness 2)** the paper provides little discussion of computational overhead, training time comparisons, or memory requirements.
>
>
> To address the concerns about computational overhead, we report the performance of each baseline at the epoch where it reaches a similar aesthetic score to that of SQDF. In the case of DDPO, we use the highest-performing epoch instead. GPU hours are measured on 2 x RTX 4090 24GB GPUs, and the peak GPU memory requirement is measured on a single RTX 4090 with batch size 1.
>
> **Table A. Comparison of computational overhead of baselines.**
> | Method-(epochs)           | Aesthetic (↑) | LPIPS-Div (↑) | ImageReward (↑) | GPU hours | Peak memory (GB) |
> |:----------:|:-------------:|:-----------:|-----------------|-----------| --|
> | DDPO-500   | 6.82          | 0.44        | -0.44           | 72.8    | 16.80
> | DDPO+KL-400  | 6.93          | 0.47        | 0.47            |  69.1         | 18.43|
> |ReFL-1160|7.86|0.45|0.54| 50.3| 8.26|
> |DRaFT-60|7.90|0.45|-1.62| 1.7| 7.80|
> | DRaFT+KL-2000   | 7.80          | 0.52        | -1.50            | 220.0 | 9.44
> | SQDF-2000 | 7.87      | 0.56    | 1.14        | 68.8     | 13.47|
>
> While SQDF requires more GPU hours than DRaFT and ReFL, direct backpropagation methods suffer from severe over-optimization as shown in their low alignment and diversity scores. In the case of DDPO, it underperforms SQDF with respect to all three metrics, even though DDPO requires more GPU hours.
>
> Meanwhile, DDPO+KL and DRaFT+KL require more GPU time than SQDF, yet still fail to mitigate over-optimization, indicating that a naive addition of KL regularization is not an effective solution.
>
> These results indicate that SQDF effectively optimizes the target reward while mitigating reward over-optimization within a reasonable amount of time.

---

### Official Review · Reviewer_TRJT · 2025-11-01

**Soundness:** 3
**Presentation:** 3
**Contribution:** 3
**Rating:** 6
**Confidence:** 4

**Summary:**

This paper focuses on fine-tuning diffusion models with downstream objectives while mitigating reward over-optimization. The proposed method solves KL-regularized off-policy RL with stabilized value estimation for controlling reward-diversity tradeoff. The method is compared with prior fine-tuning works on image generation with various rewards.

**Strengths:**

1. Reward over-optimization has been a well-known problem in diffusion fine-tuning, and the paper proposes a well-structured method with results to support their claims.
2. Each component of the method is well-motivated, and evidences are provided through ablation studies.
3. Both quantitative and qualitative results show better reward-alignment / reward-diversity compared to prior works.

**Weaknesses:**

1. Lack of fine-tuning baseline: [1] have already proposed RL fine-tuning that focuses on mitigating over-optimization.
2. Lack of training-free baselines: [2] has shown that training-free methods can mitigate over-optimization compared to fine-tuning, and [3], [4] evidence soft value function can also be used for training-free methods.
Without comparison with these methods or justification statements, it's unclear why fine-tuning is necessary.

[1] Zhang, Ziyi, et al. "Confronting reward overoptimization for diffusion models: A perspective of inductive and primacy biases." arXiv preprint arXiv:2402.08552 (2024).
[2] Kim, Sunwoo, Minkyu Kim, and Dongmin Park. "Test-time Alignment of Diffusion Models without Reward Over-optimization." The Thirteenth International Conference on Learning Representations.
[3] Li, Xiner, et al. "Derivative-free guidance in continuous and discrete diffusion models with soft value-based decoding." arXiv preprint arXiv:2408.08252 (2024).
[4] Uehara, Masatoshi, et al. "Inference-time alignment in diffusion models with reward-guided generation: Tutorial and review." arXiv preprint arXiv:2501.09685 (2025).

**Questions:**

1. There is no 'Q-learning' in the method since it approximates the value function using a consistency model. The name of the method (Soft Q-learning for Diffusion Finetuning) may be misleading.
2. Using a frozen consistency model can introduce a different type of approximation bias due to the mismatch between the posterior mean of the fine-tuned diffusion and the consistency model. Why not fine-tune the consistency model together?
3. What's the purpose of the online black-box optimization experiment? The paper lacks an explanation of which aspect of SQDF this experiment is trying to demonstrate.

---

> ### Author Response · Authors · 2025-11-20
>
> We are sincerely thankful to the reviewer for carefully identifying this weakness, which helped us refine our work.
>
> > **(Weakness 1)** Lack of fine-tuning baseline: TDPO [1] have already proposed RL fine-tuning that focuses on mitigating over-optimization.
>
> As you pointed out, TDPO proposes an RL-based approach to mitigate over-optimization. We reproduce TDPO following the authors’ official implementation, adjusting the batch size to 64 for consistency with our setting. As shown in Table 1 below, TDPO still suffers from over-optimization, even though its aesthetic score does not exceed 7. While TDPO attempts to address over-optimization, its objective function is not explicitly regularized, known to be prone to over-optimization [2][3].
>
> > **(Weakness 2)** Lack of training-free baselines: [4] has shown that training-free methods can mitigate over-optimization compared to fine-tuning, and [5], [6] evidence soft value function can also be used for training-free methods. Without comparison with these methods or justification statements, it's unclear why fine-tuning is necessary.
>
> For analyzing SQDF against training-free baselines, we include two representative methods: SVDD [5] and DAS [4]. As shown in Table A, SQDF achieves the highest aesthetic and ImageReward scores among the two training-free baselines. Although SVDD and DAS exhibit relatively high diversity, they rely on computationally expensive inference-time search procedures. In particular, SVDD and DAS require approximately 160 and 120 seconds per sample, respectively, while SQDF generates a sample in about 3 seconds, offering comparable or superior performance with lower inference cost.
>
>
> **Table A. Comparisons with additional baselines.**
> |                   | Aesthetic (↑) | LPIPS-Div (↑) | ImageReward (↑) |
> |-------------------|---------------|-------------|-----------------|
> TDPO | 6.78 | 0.39 | 0.51 |
> SVDD | 6.34 | **0.65** | 0.99 |
> DAS | 7.22 | **0.65** | 1.07 |
> | SQDF           | **7.87**          | 0.56        | **1.14**            |
>
> [1] Zhang, Ziyi, et al. "Confronting reward overoptimization for diffusion models: A perspective of inductive and primacy biases." arXiv preprint arXiv:2402.08552 (2024).
>
> [2] Black, Kevin, et al. "Training Diffusion Models with Reinforcement Learning." The Twelfth International Conference on Learning Representations.
>
> [3] Fan, Jiajun, et al. "Online reward-weighted fine-tuning of flow matching with wasserstein regularization." The Thirteenth International Conference on Learning Representations. 2025.
>
> [4] Sunwoo Kim, Minkyu Kim, and Dongmin Park. "Test-time alignment of diffusion models without reward over-optimization." arXiv preprint arXiv:2501.05803 (2025).
>
> [5] Li, Xiner, et al. "Derivative-free guidance in continuous and discrete diffusion models with soft value-based decoding." arXiv preprint arXiv:2408.08252 (2024).
>
> [6] Uehara, Masatoshi, et al. "Inference-time alignment in diffusion models with reward-guided generation: Tutorial and review." arXiv preprint arXiv:2501.09685 (2025).

---

> ### Author Response · Authors · 2025-11-20
>
> We sincerely thank the reviewer for raising thoughtful questions about our work.
>
> >**(Question 1)**  There is no 'Q-learning' in the method since it approximates the value function using a consistency model. The name of the method (Soft Q-learning for Diffusion Finetuning) may be misleading.
>
> As you pointed out, we do not train Q-function. Accordingly, we have revised the name of our method from Soft "Q-learning" Diffusion Finetuning to Soft "Q-based" Diffusion Finetuning to more accurately reflect our approach. Thank you for the feedback.
>
> > **(Question 2)** Using a frozen consistency model can introduce a different type of approximation bias due to the mismatch between the posterior mean of the fine-tuned diffusion and the consistency model. Why not fine-tune the consistency model together?
>
> Our KL-regularized RL framework relies on the following soft Q-function approximation:
>
> $$
> \quad Q^*\_{\text{soft}}(x_t,x_{t-1})= \alpha \log \mathbb{E}\_{x_0,...,x_{t-2}\sim p^{\prime}(\cdot|x_{t-1})}\left[\exp\left(\frac{r(x_0)}{\alpha}\right)\Big|x_{t-1}\right].
> $$
>
> Please note that the expectation is taken with respect to the pretrained model $p'$. In prior works [1][2], this expectation is commonly approximated via Tweedie’s formula as follows:
>
> $$
> \quad Q^*\_{\text{soft}}(x_t,x_{t-1}) \approx r(\hat x_0(x_{t-1})) \quad \hat{x}\_0(x_{t-1})=\mathbb{E}\_{p'}[x_0|x_{t-1}].
> $$
>
> However, Tweedie’s formula becomes inaccurate at high noise levels. SQDF addresses this limitation by replacing this estimate with a Consistency Model, which yields a more accurate and stable approximation across all noise levels.
>
> To this end, the consistency model must act as a static replacement for the expectation under the pre-trained diffusion model $p'$, not the current diffusion model $p_{\theta}$. Jointly training the CM contradicts this requirement. Therefore, the consistency model must remain frozen, serving as a stable approximation of the original reference policy's posterior mean.
>
> [1] Hyungjin Chung, Jeongsol Kim, Michael Thompson Mccann, Marc Louis Klasky, and Jong Chul Ye. Diffusion posterior sampling for general noisy inverse problems. In The Eleventh International Conference on Learning Representations, 2023.
>
> [2] Li, Xiner, et al. "Derivative-free guidance in continuous and discrete diffusion models with soft value-based decoding." arXiv preprint arXiv:2408.08252 (2024).
>
> > **(Question 3)** What's the purpose of the online black-box optimization experiment? The paper lacks an explanation of which aspect of SQDF this experiment is trying to demonstrate.
>
> There is a pressing need to optimize black-box reward functions with high sample efficiency, as reward calls are often expensive [1]. Recently, pre-trained generative models are often used as a starting point of the iterative online black-box optimization, since they can effectively narrow down the initial search space to the pre-trained data manifold [2][3]. Preserving sample diversity and naturalness is crucial in this setting; a loss of diversity and naturalness leads to inefficient exploration and thus poor sample efficiency [4]. To address these needs, we evaluate SQDF's capability to effectively optimize the black-box reward and mitigate overoptimization, while achieving high sample efficiency. As shown in the results in section 5.2, SQDF outperforms baselines under the same oracle budget with higher diversity and alignment scores.
>
> [1] Gao, Wenhao, et al. "Sample efficiency matters: a benchmark for practical molecular optimization." Advances in neural information processing systems 35 (2022): 21342-21357.
>
> [2] Uehara, Masatoshi, et al. "Feedback Efficient Online Fine-Tuning of Diffusion Models." International Conference on Machine Learning. PMLR, 2024.
>
> [3] Zhou, Xiangxin, Liang Wang, and Yichi Zhou. "Stabilizing policy gradients for stochastic differential equations via consistency with perturbation process." Proceedings of the 41st International Conference on Machine Learning. 2024.
>
> [4] Bengio, Emmanuel, et al. "Flow network based generative models for non-iterative diverse candidate generation." Advances in neural information processing systems 34 (2021): 27381-27394.

---

### Official Review · Reviewer_M6wy · 2025-11-02

**Soundness:** 3
**Presentation:** 4
**Contribution:** 4
**Rating:** 8
**Confidence:** 3

**Summary:**

- Problem: when you RL diffusion models it overoptimizes the reward and mode collapses and stops producing coherent images.
- Solution:
  - KL regularization to the original model (also common in past work)
  - Soft-Q function (this is a normal Q-fn with an entropy term)
  - Discount factor for better credit assignment
  - Consistency models for more stable RL training
  - Off-policy replay buffer - prevent mode collapse
Algorithm works like this:
  - Generate samples from diffusion model, store them in the replay buffle
  - Sample one of the partially noised samples, denoise it 1 step.
  - Then take the consistency model and denoise the rest of the way to a clean image.
  - Run the clean image through the RM.
  - Backprop using policy gradient.
Experiments
  - 3 things they want to show (a) still good samples (b) less semantic collapse (c) less diversity collapse.
  - To measure semantic, they use pretrained prompt-image alignment models like ImageReward.
  - To measure diversity, they look at the mean pairwise distance as computed with 2 measures: LPIPS and Dreamsim features cosine similarity.
  - Their experimental results look pretty legit:
     - Show there’s an alignment-diversity tradeoff, but their method expands the pareto frontier
     - Shows that there is an aesthetic-alignment tradeoff, but they minimize and ~almost remove it. They are slightly better than ReFL and much better than the other methods.
     - Shows there is an aesthetic-diversity tradeoff and they’re way better than other methods.
  - Ablations show the discount helps a bunch. It looks like the buffer helps with diversity (but slightly hurts reward) and the CM helps with reward (but slightly hurts diversity). But maybe these differences are small enough it’s noise?

**Strengths:**

- Substantial baselines - they compared against several past works. They tested whether a simple addition to baselines (adding KL regularization) would improve them, and found that their method was still superior.
- Ablations - removed each component they added to confirm that it contributes to the final result.
- Clarity - the paper is clear and easy to read. The diagrams make sense.
Significant improvement above prior work, which is visible both in the qualitative results and the graphs.

**Weaknesses:**

See “questions” section for more details.

It is not clear to me that the minor improvements in diversity produced by the buffer (.01 to .02 in Table 2) is worth the extra complexity adding it to the algorithm requires and the hit to aesthestic score.

Similarly it seems like without the consistency model diversity improves (though this is a minor effect, the improvements to aesthetic score I think suggest the CM is still a good contribution to the algorithm).

**Questions:**

- In Fig 3, how did you get the diff points on each curve?
- Table 1 - what is the number in parens. (confidence interval? standard deviation?)
- Are the qualitative results randomly sampled or cherry-picked?
- I suggest moving Algorithm 1 to the main paper, it was very helpful in understanding the algorithm.

---

> ### Author Response · Authors · 2025-11-20
>
> >**(Weakness 1)** It is not clear to me that the minor improvements in diversity produced by the buffer (.01 to .02 in Table 2) is worth the extra complexity adding it to the algorithm requires and the hit to aesthestic score.
>
> We appreciate the reviewer's feedback regarding the effect of the buffer. We agree with the reviewer that the numerical gains in diversity might seem marginal. However, considering the narrow range of LPIPS (0.45 ~ 0.65) and Dreamsim (0.35 ~ 0.75) in our experiments, the improvement by buffer is about 5% which is non-trivial.
>
> Furthermore, we would like to clarify that the buffer could be understood as a technique to explicitly manage the reward-diversity trade-off. To support this argument, we additionally conduct an ablation study of the buffer in an online black-box optimization setting.
>
> **Table A. Results of online black-box optimization**
> | Method    | Aesthetic (↑) | LPIPS-Div (↑) | Dreamsim-Div (↑) |
> |---------------|----|-----|----|
> | SQDF w/o Buffer     |**7.81**|0.53|0.5|
> | SQDF w Buffer |7.64|**0.56**|**0.54**|
>
> Uniform replay buffer consistently enhances the diversity in both Text-to-Image generation and online black-box optimization tasks at the cost of reward (See Table 2 in Section 5.3 and Table A above).
>
>
> >**(Weakness 2)** Similarly it seems like without the consistency model diversity improves (though this is a minor effect, the improvements to aesthetic score I think suggest the CM is still a good contribution to the algorithm).
>
> As shown in Table 2 in the paper, the aesthetic score of SQDF w/o CM is noticeably lower. In such cases, the higher diversity may simply result from the lower target reward level, rather than indicating an actual advantage.
>
> To show that the consistency model improves reward optimization without compromising diversity, we additionally conduct an experiment where we adjust the KL regularization coefficient $\alpha$ for both SQDF w/o CM and SQDF, matching them to similar target reward levels.
>
> **Table B. Comparison of SQDF and SQDF w/o CM at similar target reward level ($\approx 7.8$)**
> | Method    | $\alpha$ | Aesthetic (↑) |  DreamSim-Div (↑) |
> |---------------|----|------------|-----------|
> | SQDF w/o CM   | 1.5|7.73|0.56|
> | SQDF | 2.0 | **7.87**|**0.58**|
>
> **Table C. Comparison of SQDF and SQDF w/o CM at similar target reward level ($\approx 7.1$)**
> | Method    | $\alpha$ | Aesthetic (↑) |  DreamSim-Div (↑) |
> |---------------|----|------------|-----------|
> | SQDF w/o CM      |  2.0 | 7.10|0.62 |
> | SQDF | 3.0 | **7.16**|**0.64** |
>
> The result shows that SQDF achieves higher diversity at a comparable target reward, supporting the interpretation that CM improves reward optimization without compromising diversity. This improvement results from the consistency model producing a more accurate posterior mean estimate, which in turn provides a more reliable reward signal.
>
>
> > **(Question 1)** In Fig 3, how did you get the diff points on each curve?
>
> For the aesthetic score, we set the terminal epoch of the SQDF, DRaFT, and ReFL at which each method first reached a target reward of 8.0. In the case of DDPO, we set the terminal epoch as the point with maximum reward before collapse, since it can't reach 8.0. For the HPS, we set the terminal epoch as 500 for all methods. Since DDPO shows severe diversity collapse, we set its terminal epoch to 200. Once the terminal epoch is defined for each method, we plot the curve by selecting 6 epochs at uniform intervals between the start and this terminal epoch. Please refer to Appendix D.2.2 for more details.
>
>
> > **(Question 2)** Table 1 - what is the number in parens. (confidence interval? standard deviation?)
>
> Thank you for pointing out the missing clarification. The values in parentheses of Table 1 represent the standard deviation of 3 random seeds. We will update the caption for Table 1 to explicitly state this.
>
> > **(Question 3)** Are the qualitative results randomly sampled or cherry-picked?
>
> Thank you for raising this question, and we apologize for not clearly explaining how the qualitative results are selected. In the main qualitative figures, we select samples from SQDF and then report the corresponding samples at the same index for all other methods.
>
> To remove any ambiguity, we additionally provide uncurated samples in Appendix F, showing seven consecutive samples from the evaluation set without any selection. As shown in the figures in Appendix F, these uncurated results exhibit similar trends, confirming that SQDF’s qualitative improvements are not dependent on sample selection.
>
>
> > **(Question4)** I suggest moving Algorithm 1 to the main paper, it was very helpful in understanding the algorithm.
>
> We are glad that Algorithm 1 in the appendix helped clarify your understanding of our method. After the discussion period ends, we will move Algorithm 1 to the main section if we have enough space after incorporating the revisions.

---

### Author Response · Authors · 2025-11-20

We sincerely thank the review committee for their thoughtful and constructive feedback. We appreciate the recognition of our paper’s strengths, consistently emphasized across reviewers: **Strong empirical results** (M6wy, TRJT, 3B3B, 5VJm), **Thorough ablations** (M6wy, TRJT, 3B3B, 5VJm), **Important problem setting** (TRJT, 5VJm), **Substantial baselines** (M6wy, 3B3B), and **Clarity of presentation** (M6wy, 5VJm)

In response to the reviewers' feedback, we provide a brief summary of the additional major experiments and modifications:

---
### Additional Experiments

**Expanding SQDF to more advanced diffusion models**

- We evaluate SQDF on SDXL, confirming that SQDF reliably optimizes the target reward while mitigating over-optimization across different backbone sizes.


**Additional baselines**

- We add both a fine-tuning baseline (TDPO) and training-free baselines (SVDD, DAS), and show that SQDF achieves consistently better performance.

---

### Revisions

- Section 4.1: Expand explanation of the reparameterized policy gradient.
- Figure 3: Add DDPO as a baseline to the HPS optimization task.
- Appendix F: Extend the evaluation to advanced backbones(SDXL).
- Appendix G: Add uncurated samples for transparency.
- Algorithm 1: Correct minor errors and improve clarity.

---

### Source code release

We release our source code of SQDF for full reproducibility at https://anonymous.4open.science/r/SQDF-B66C.

---

We thank the reviewers again for their constructive feedback and hope that the revisions and additional results sufficiently address all concerns.

---

### Author Response · Authors · 2025-11-28

Dear Reviewers,

We hope this message finds you well.

We have noticed that reviewers have not yet participated in the discussion. As the author-reviewer discussion period is set to conclude in a few days, we kindly wish to remind you of the opportunity to provide your valuable feedback.

Your insights and perspectives are incredibly important to us and will greatly contribute to improving our work. If you have any questions or require further information, please do not hesitate to reach out.

We sincerely appreciate your time and consideration, and we look forward to your input.

Best regards, The Authors

---

### Comment · Area_Chair_Hwvg · 2025-11-28
**Please Check the Authors' Responses**

Dear Reviewers,

The authors have posted their responses. Could you please take a moment to review their responses and check whether your concerns have been adequately addressed (if you have not done it yet)? If possible, kindly initiate the discussion at your earliest convenience.

Your timely assistance is essential for keeping the review process on track. Thank you very much for your support and contribution.

Best regards, Your AC

---

### Meta-Review · Area_Chair_YUpg · 2026-01-07

**Summary:**

The paper initially received one accept, two borderline accepts, and one borderline reject. The reviewers consistently recognized the strong empirical performance of the proposed method (M6wy, TRJT, 3B3B, 5VJm), supported by thorough ablation studies (M6wy, TRJT, 3B3B, 5VJm). They also highlighted the importance of the problem setting (TRJT, 5VJm), the substantial baseline comparisons (M6wy, 3B3B), and the clarity of the presentation (M6wy, 5VJm). In the rebuttal, the authors further strengthened the paper by adding additional experiments on more advanced diffusion models, including evaluations of SQDF on SDXL, incorporating more baseline comparisons (TDPO, SVDD, DAS), and providing clearer explanations of the reparameterized policy gradient formulation. The authors also provided detailed, point-by-point responses to all reviewer concerns, particularly addressing the issues raised by Reviewer 5VJm, who initially gave a borderline reject. Overall, the rebuttal satisfactorily resolves all major concerns. We therefore recommend acceptance of the paper, with the expectation that all discussed additions and clarifications are fully incorporated into the final manuscript.

**Reviewer Concerns:**

I think all the concerns of reviewers are properly addressed by the authors.

**Reviewer Scores:**

Three of four reviewers give positive ratings to the paper: one accept and two borderline accept. For all the reviewers, the authors give a very detailed rebuttal to each weakness and question to each reviewer. I think under this situation, all other reviewers who give positive ratings would keep their scores or upgrade the scores. For Reviewer 5VJm who gave borderline reject, I think he will upgrade his score as well if he is able to participate fully in the discussion.

---

### Decision · Program_Chairs · 2026-01-26

Accept (Poster)